# Verified SHAP:
# Provable Bounds for Exact Shapley Values of Neural Networks

David Boetius [* 1]   Shahaf Bassan [* 2]   Guy Katz [2]   Stefan Leue [1]   Tobias Sutter [3]

## Abstract

Shapley additive explanations (SHAP) are widely recognised as computationally intractable for neural networks, since they induce an exponential search space over the input features. In this work, we take a first step towards scaling exact SHAP computation to larger search spaces by introducing an algorithm that leverages recent advances in neural network verification to compute arbitrarily tight exact lower and upper bounds on SHAP values for neural networks, ultimately recovering the exact SHAP values. We demonstrate that our approach scales to orders of magnitude larger search spaces than state-of-the-art exact methods. This provides an important first step towards exact SHAP computation and establishes a principled cornerstone for evaluating statistical approximation methods on larger search spaces.

## 1. Introduction

Shapley additive explanations (SHAP) is a widely used post-hoc explainability method for attributing a machine learning (ML) model's predictions to its input features. However, a central limitation of SHAP is its high computational complexity. While *exact* SHAP values can be computed efficiently for certain model classes, such as tree-based (Lundberg et al., 2020; Mitchell et al., 2022b; Yang, 2021) and additive models (Enouen & Liu, 2025), they unfortunately become prohibitively expensive to compute for neural networks, where explanations are often most needed. Concretely, exact SHAP computation induces an *exponentially* large search space over feature subsets, rendering exact computation impractical for most neural networks.

Consequently, the literature primarily focuses on *estimating* SHAP values of neural networks, using methods such as KERNELSHAP (Lundberg & Lee, 2017; Covert & Lee, 2021), LEVERAGESHAP (Musco & Witter, 2025), TREEMSR (Witter et al., 2025), FASTSHAP (Jethani et al., 2022), and DEEPSHAP (Chen et al., 2020). However, these methods share two fundamental limitations:

*(i)* by construction, they only provide *approximations* of the true SHAP values, which can be inaccurate in challenging settings such as highly non-linear models or scenarios with strong feature interactions, and

*(ii)* they lack a principled *evaluation* framework: since computing 'ground-truth' exact SHAP values for neural networks is prohibitively expensive, estimators are evaluated on toy-sized neural networks or ML models where exact SHAP is feasible—both of which are settings that may not generalise to real-world neural networks.

**Our Contributions.** In this work, we present *Verified SHAP* (VERISHAP), the first algorithm to leverage recent advances in *neural network verification* (Wang et al., 2021; Zhou et al., 2024; Brix et al., 2024; Kotha et al., 2023; Ferrari et al., 2022; Zhou et al., 2025) to compute *exact* SHAP values and *provable bounds* on exact SHAP values for neural networks, scaling to significantly larger search spaces than previously tractable by exact methods. Our contribution serves three complementary purposes:

*(i)* it represents a first step towards scaling the exact computation of SHAP values for neural networks to larger search spaces, with the promise that continued progress in neural network verification will enable our approach to scale to yet larger search spaces;

*(ii)* it allows computing provable SHAP lower and upper bounds up to *arbitrary precision*, enabling fast, yet meaningful and trustworthy explanatory insights; and

*(iii)* it provides a principled '*ground-truth*' explanation baseline for evaluating statistical approximation methods on larger search spaces, serving as an important cornerstone for SHAP evaluation in the explainable AI literature.

---

[*]Equal contribution  [1]University of Konstanz, Konstanz, Germany [2]Hebrew University of Jerusalem, Jerusalem, Israel [3]University of St.Gallen, St.Gallen, Switzerland. Correspondence to: David Boetius <david.boetius@uni-konstanz.de>, Shahaf Bassan <shahaf.bassan@gmail.com>.

*Proceedings of the $43^{rd}$ International Conference on Machine Learning*, Seoul, South Korea. PMLR 306, 2026. Copyright 2026 by the author(s).

**From Neural Network Verification to Exact SHAP.** The problem of automatically verifying that a neural network satisfies specifications, such as adversarial robustness, has seen rapid progress in recent years, driven largely by branch-and-bound verification methods (Bunel et al., 2020; Wang et al., 2021). Recent works (e.g., Wu et al., 2023; Bassan et al., 2025; Izza et al., 2024) adapt these techniques to compute certain forms of explanations, but instead of targeting the widely used SHAP framework, they focus on robustness-based explanations that are well-suited for applying existing neural network verifiers. In contrast, SHAP values present significant challenges for applying neural network verifiers, as SHAP is *(i)* inherently *discrete*; *(ii) probabilistic*, since SHAP is defined via expectations; and *(iii)* involves a summation over an *exponential* number of feature sets. In this work, we present the first algorithmic framework that overcomes these challenges and enables computing exact SHAP values using techniques from neural network verification.

**Our Algorithm.** Our VERISHAP algorithm is based on an incremental branch-and-bound neural network verification approach. VERISHAP recursively partitions the feature space, computing upper and lower bounds on the SHAP value within each resulting region. The central challenges lie in deriving sufficiently tight bounds and in designing efficient splitting strategies. Conceptually, our approach can be viewed as exploiting a decomposition of the neural network's input space into near-linear regions, within which the model's output—and consequently the SHAP value—can be tightly bounded. Owing to the recursive structure of the branch-and-bound procedure, our algorithm can produce arbitrarily tight upper and lower bounds on the SHAP values of a model $f$. Eventually, it recovers the exact SHAP values.

**Empirical Evaluation.** We conduct an extensive evaluation of our framework on both tabular and vision benchmarks, demonstrating that it scales to substantially larger search spaces than current state-of-the-art exact SHAP approaches. Furthermore, we compare our exact SHAP bounds with a broad range of statistical SHAP estimators, showing that our algorithm complements these estimators when high accuracy and precision are required. Finally, we use VERISHAP to evaluate SHAP estimators on neural networks for larger search spaces than were previously feasible.

Overall, our results highlight the promise of VERISHAP as both a first step towards exact SHAP computation for neural networks and a central benchmark for evaluating SHAP estimators. Due to space constraints, we only include brief proof sketches for our theoretical results in the main paper and defer the complete proofs to Appendix A.

## 2. Preliminaries

This section formally defines SHAP values and introduces bound propagation as used for neural network verification.

**Notation.** Let $S$ be a set and $n \in \mathbb{N}$. We define $[n] := \{1, \ldots, n\}$, denote the cardinality of $S$ as $|S|$, the power set of $S$ as $\mathcal{P}(S)$, and the all-zero and all-one vectors as $\mathbf{0}_n, \mathbf{1}_n \in \mathbb{R}^n$, respectively. Calligraphic upper-case letters like $\mathcal{S} \subseteq \mathcal{P}(S)$ denote sets of subsets, while sans-serif upper-case letters like $\mathsf{S} \subseteq \mathcal{P}(\mathcal{P}(S))$ denote families of sets of subsets. For $\mathbf{x}, \mathbf{z} \in \mathbb{R}^n$ and $S \subseteq [n]$, we define $(\mathbf{x}_S; \mathbf{z}_{\bar{S}}) := \mathbf{u}$, where $\mathbf{u}_i = \mathbf{x}_i$ if $i \in S$ and $\mathbf{u}_i = \mathbf{z}_i$ if $i \in \bar{S} = [n] \setminus S$ for $i \in [n]$. Given $\underline{\mathbf{x}}, \overline{\mathbf{x}} \in \mathbb{R}^n$ with $\underline{\mathbf{x}} \leq \overline{\mathbf{x}}$, $[\underline{\mathbf{x}}, \overline{\mathbf{x}}]$ denotes the hyper-box $\{\mathbf{x} \in \mathbb{R}^n \mid \underline{\mathbf{x}} \leq \mathbf{x} \leq \overline{\mathbf{x}}\}$.

### 2.1. Shapley Additive Explanations (SHAP)

Shapley Additive Explanations (SHAP) are a popular game-theoretic explainability technique for attributing an ML model's output to input features using the classic Shapley value framework. Given a black-box model $f : \mathbb{R}^n \to \mathbb{R}^m$ (in our case, a neural network), and some specific input $\mathbf{x} \in \mathbb{R}^n$, we wish to locally explain the prediction $f(\mathbf{x})$. Formally, the SHAP value attribution of feature $i \in [n]$ is

$$\varphi_i := \sum_{S \in \mathcal{S}_i} \frac{|S|!(n - |S| - 1)!}{n!}(v(S \cup \{i\}) - v(S))$$

$$= \sum_{S \in \mathcal{S}_i} \lambda(|S|)\Delta_i(S), \tag{1}$$

where $\mathcal{S}_i := \mathcal{P}([n] \setminus \{i\})$, $\lambda(k) := \frac{1}{n}\binom{n-1}{k}^{-1}$ is the *weight* of a *coalition* $S \in \mathcal{S}_i$ of size $|S| = k \in \{0, \ldots, n-1\}$, $v(S)$ is the *value function*, measuring the contribution of $S$, and

$$\Delta_i(S) := v(S \cup \{i\}) - v(S) \tag{2}$$

is the *marginal contribution* of $i$ to the coalition $S$. A central question that arises when using SHAP values is the definition of the value function $v(S)$ (Sundararajan & Najmi, 2020). In this work, we use the standard *marginal* value function $v(S) := \mathbb{E}_{\mathbf{z} \sim \mathcal{D}}[f(\mathbf{x}_S; \mathbf{z}_{\bar{S}})]$, where $(\mathbf{x}_S; \mathbf{z}_{\bar{S}})$ denotes a vector where the features in $S$ are taken from the vector $\mathbf{x}$ and those in $\bar{S} = [n] \setminus S$ are taken from the vector $\mathbf{z}$. Moreover, $\mathcal{D}$ denotes the data distribution, which is typically a marginalisation over a background dataset sampled from the training dataset.

Computing $\varphi_i$ exactly is #P-hard (Van den Broeck et al., 2022), a class viewed as even 'harder' than NP-hard (Arora & Barak, 2009). However, this does not determine whether $\varphi_i$ may sometimes be efficiently computable in practice.

### 2.2. Neural Network Verification

Neural network verification seeks to automatically prove input-output properties of neural networks over computa-

tionally hard (e.g., NP-hard) problems. Given a neural network $f : \mathbb{R}^n \to \mathbb{R}^m$, an input specification $\psi_{\text{in}}$ (e.g., interval bounds on input features), and an output specification $\psi_{\text{out}}(f(\mathbf{x}))$, a neural network verifier formally proves whether the output specification holds for all inputs satisfying the input specification.

**Bound Propagation.** Bound propagation is a key ingredient in state-of-the-art neural network verification methods. It operates by deriving lower and upper bounds $\underline{f}, \overline{f}$ on the output of a neural network $f$ over an input hyper-box $X := [\underline{\mathbf{x}}, \overline{\mathbf{x}}]$. These bounds can then be used to prove whether the output specification $\psi_{out}(f(\mathbf{x}))$ holds for all $\mathbf{x} \in X$. Formally, the bounds satisfy $\underline{f} \leq f(\mathbf{x}) \leq \overline{f}, \forall \mathbf{x} \in X$. The bound propagation methods we consider here also satisfy $\underline{f} = \overline{f}$ if $\underline{\mathbf{x}} = \overline{\mathbf{x}}$ (Moore et al., 2009; Boetius et al., 2025).

Bound propagation relies on the fact that neural networks can be defined as a composition of more fundamental functions $f = f^{(K)} \circ \cdots \circ f^{(1)}$, where $f^{(1)}, \ldots, f^{(K)}$ are the layers of the network. The input bounds $X$ are propagated layer by layer through the network, where each step obtains an outer approximation of the layer's function image. This is done until the output layer, where $[\underline{f}, \overline{f}]$ is obtained.

**Interval Bound Propagation.** One of the simplest bound propagation techniques is interval bound propagation (IBP) (Moore et al., 2009), also known as interval arithmetic, which is based on a set of rules for computing bounds for each layer type. For example, if $f^{(1)}(\mathbf{x}) = \mathbf{W}\mathbf{x} + \mathbf{b}$,

$$\underline{\mathbf{z}} = \max(0, \mathbf{W})\underline{\mathbf{x}} + \min(0, \mathbf{W})\overline{\mathbf{x}} + \mathbf{b} \leq f^{(1)}(\mathbf{x})$$
$$\overline{\mathbf{z}} = \max(0, \mathbf{W})\overline{\mathbf{x}} + \min(0, \mathbf{W})\underline{\mathbf{x}} + \mathbf{b} \geq f^{(1)}(\mathbf{x}),$$

provides us with bounds on $f^{(1)}(\mathbf{x})$ for $\mathbf{x} \in X = [\underline{\mathbf{x}}, \overline{\mathbf{x}}]$, where $\min$ and $\max$ are applied element-wise. We can now propagate $[\underline{\mathbf{z}}, \overline{\mathbf{z}}]$ forward to compute bounds on the next layer. For example, if $f^{(2)}(\mathbf{z}) = \max(0, \mathbf{z})$, we can use $\underline{\mathbf{z}}' = \max(0, \underline{\mathbf{z}}) \leq \max(0, \mathbf{z}) \leq \max(0, \overline{\mathbf{z}}) = \overline{\mathbf{z}}'$ to obtain bounds on $(f^{(2)} \circ f^{(1)})(\mathbf{x})$. Repeating this propagation for $f^{(3)}$ with $[\underline{\mathbf{z}}', \overline{\mathbf{z}}']$ and, subsequently, the remaining $f^{(k)}$, eventually yields bounds on $f(\mathbf{x})$ for $\mathbf{x} \in X$.

**Linear Bound Propagation.** While IBP propagates intervals from inputs to outputs, linear bound propagation (LBP) approaches (Zhang et al., 2018; Wang et al., 2018b; Singh et al., 2019; Xu et al., 2021) propagate two linear functions from outputs to inputs through $f$. While the linear functions are eventually converted to an interval $[\underline{f}, \overline{f}]$, LBP can significantly reduce the *approximation error* $\overline{f} - \max_{\mathbf{x} \in X} f(\mathbf{x})$ and $\min_{\mathbf{x} \in X} f(\mathbf{x}) - \underline{f}$ that arises both in IBP and LBP. In particular, LBP introduces no approximation error for compositions of linear functions. We refer to Zhang et al. (2018) for a detailed introduction to the LBP method CROWN.

## 3. Provable Bounds on Exact SHAP

In this section, we present our approach for computing exact SHAP values of neural networks. Unlike linear models and decision trees, existing exact methods for neural networks rely on enumerating all coalitions $S \in \mathcal{S}_i$ to evaluate Equation (1). Our method instead iteratively refines a partition of $\mathcal{S}_i$ to compute guaranteed bounds $\underline{\varphi_i} \leq \varphi_i \leq \overline{\varphi_i}$ using bound propagation. This allows us to compute tight bounds on $\varphi_i$ to any desired precision.

We now introduce our approach in more detail, starting with computing guaranteed SHAP bounds based on a partition. Next, we discuss our partitioning scheme and prove that our scheme terminates after a finite number of iterations. Finally, we show that our algorithm terminates immediately for linear models and can terminate early for piecewise-linear models, such as ReLU-activated neural networks. For clarity, we call the elements of a partition *branches* in line with common branch-and-bound terminology.

**Bounds on $\varphi_i$.** Let $\mathsf{B}^{(t)} := (\mathcal{B}_1^{(t)}, \ldots, \mathcal{B}_K^{(t)})$ be a partition of $\mathcal{S}_i$, so that $\mathcal{B}_k^{(t)} \subseteq \mathcal{S}_i, \forall k \in [K]$. For each branch $\mathcal{B} \in \mathsf{B}^{(t)}$, we can compute lower and upper bounds $[\underline{\Delta_{i\mathcal{B}}}, \overline{\Delta_{i\mathcal{B}}}]$, such that $\underline{\Delta_{i\mathcal{B}}} \leq \Delta_i(S) \leq \overline{\Delta_{i\mathcal{B}}}$ for all $S \in \mathcal{B}$, using bound propagation for neural networks on $\Delta_i(S)$ as defined in Equation (2). The practical aspects of this are discussed in Section 4. Assuming we can also compute $\Lambda_{\mathcal{B}} := \sum_{S \in \mathcal{B}} \lambda(|S|)$, we obtain the result below.

**Theorem 3.1** (SHAP Bounds). *Let $\mathsf{B}^{(t)}$ be a partition of $\mathcal{S}_i$. It holds that*

$$\underline{\varphi_i}^{(t)} := \sum_{\mathcal{B} \in \mathsf{B}^{(t)}} \Lambda_{\mathcal{B}}\underline{\Delta_{i\mathcal{B}}} \leq \varphi_i \leq \sum_{\mathcal{B} \in \mathsf{B}^{(t)}} \Lambda_{\mathcal{B}}\overline{\Delta_{i\mathcal{B}}} =: \overline{\varphi_i}^{(t)}$$

*Proof Sketch.* We group coalitions $S$ into branches $\mathcal{B}$. By definition, each branch's total Shapley weight $\Lambda_{\mathcal{B}}$ is exactly the sum of the coalition weights $\lambda(|S|)$ inside it. Bound propagation ensures that each coalition's contribution is bounded by the contribution bounds $[\underline{\Delta_{i_{\mathcal{B}}}}, \overline{\Delta_{i\mathcal{B}}}]$ of the respective branch. Finally, since the branches form a partition, summing $\Lambda_{\mathcal{B}}\underline{\Delta_{i_{\mathcal{B}}}}$ and $\Lambda_{\mathcal{B}}\overline{\Delta_{i\mathcal{B}}}$ provides a lower, respectively, upper bound on the Shapley value $\varphi_i$.

**Partitioning $\mathcal{S}_i$.** We now describe how we create and refine the partition $\mathsf{B}$. We define the branches $\mathcal{B} \in \mathsf{B}$ in terms of sets of *included* and *excluded features* $\mathcal{I}, \mathcal{E} \subseteq [n] \setminus \{i\}$ so that $\mathcal{B} := \{S \in \mathcal{S}_i \mid \mathcal{I} \subseteq S, \mathcal{E} \cap S = \emptyset\}$. As an initial partition, we use the trivial partition $\mathsf{B}^{(1)} := (\mathcal{B}_1^{(1)})$ where $\mathcal{B}_1^{(1)}$ is defined by $\mathcal{I}_1^{(1)} = \mathcal{E}_1^{(1)} = \emptyset$, so that $\mathcal{B}_1^{(1)} = \mathcal{S}_i$. Our refinement proceeds recursively. The $t$-th partition $\mathsf{B}^{(t)}$ is derived from the previous partition $\mathsf{B}^{(t-1)} = (\mathcal{B}_1^{(t-1)}, \ldots, \mathcal{B}_K^{(t-1)})$ by selecting a

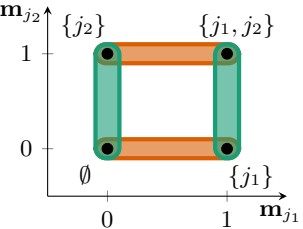 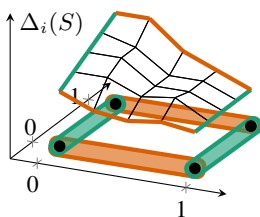

*Figure 1.* **Splitting in VERISHAP.** For $\mathcal{S}_i = \{j_1, j_2\}$, there are two options, ■ and ■, for splitting the initial partition (left). For the split ■, the branches are $\mathcal{B}_1 = \{\emptyset, \{j_2\}\}$ and $\mathcal{B}_2 = \{\{j_1\}, \{j_1, j_2\}\}$, defined by $\mathcal{I}_1 = \mathcal{E}_2 = \emptyset$ and $\mathcal{E}_1 = \mathcal{I}_2 = \{j_1\}$. This split allows computing the exact SHAP value directly since the contribution function $\Delta_i(S)$ is linear for this split (right).

branch $\mathcal{B}_k \in \mathsf{B}^{(t-1)}$ defined by $\mathcal{I}_k$ and $\mathcal{E}_k$ and selecting a feature $j \in ([n] \setminus \{i\}) \setminus (\mathcal{I}_k \cup \mathcal{E}_k)$. Intuitively, we split $\mathcal{B}_k$ by putting all coalitions $S \in \mathcal{B}_k$ containing the selected feature $j$ into the first new branch $\mathcal{B}'_k$ and putting all $S \in \mathcal{B}_k$ not containing $j$ into the second new branch $\mathcal{B}''_k$. Formally, $\mathsf{B}^{(t)} := (\mathcal{B}_1^{(t-1)}, \ldots, \mathcal{B}'_k, \mathcal{B}''_k, \ldots, \mathcal{B}_K^{(t-1)})$, where $\mathcal{B}'_k$ is defined by $\mathcal{I}'_k := \mathcal{I}_k \cup \{j\}$ and $\mathcal{E}'_k := \mathcal{E}_k$, and $\mathcal{B}''_k$ is defined by $\mathcal{I}''_k := \mathcal{I}_k$ and $\mathcal{E}''_k := \mathcal{E}_k \cup \{j\}$. Figure 1 illustrates this partitioning strategy. We discuss strategies for selecting branches and features to split in Section 5.

A crucial assumption of our bounding approach is that we can compute $\Lambda_\mathcal{B}$ efficiently. As we show next, the partitioning approach described above enables this.

**Proposition 3.2** (Closed-Form Expression for $\Lambda_\mathcal{B}$). *Consider $\mathcal{B} \in \mathsf{B}^{(t)}$ defined by $\mathcal{I}$ and $\mathcal{E}$. Letting $r := |\mathcal{I}|$ and $s := |\mathcal{I}| + |\mathcal{E}|$, we have $\Lambda_\mathcal{B} = \sum_{S \in \mathcal{B}} \lambda(|S|) = (s+1)^{-1} \binom{s}{r}^{-1}$.*

*Proof Sketch.* The proposition is established through a sequence of combinatorial manipulations, leveraging a beta-function identity and the truncated-sum binomial theorem.

Furthermore, our refinement process eventually produces a partition for which $\varphi_i$ can be computed exactly.

**Theorem 3.3** (Termination). *There exists $t \in \mathbb{N}$, such that $\underline{\varphi_i}^{(t)} = \overline{\varphi_i}^{(t)} = \varphi_i$.*

*Proof Sketch.* Since the number of features is finite, our refinement process eventually produces a partition where each branch $\mathcal{B}$ contains a single set $S \in \mathcal{S}_i$. At this point, bound propagation computes $\underline{\Delta_{i_\mathcal{B}}} = \overline{\Delta_{i_\mathcal{B}}} = \Delta_i(S)$ so that the SHAP values are computed exactly.

As a direct consequence, our algorithm can also compute SHAP values up to arbitrary precision, which, in practice, can be faster than computing the exact SHAP values.

**Corollary 3.4** (Arbitrary Precision). *Given any precision level $\delta \in \mathbb{R}_{\geq 0}$, there exists $t \in \mathbb{N}$, such that $\overline{\varphi_i}^{(t)} - \underline{\varphi_i}^{(t)} \leq \delta$. This implies $|\hat{\varphi}_i - \varphi_i| \leq \delta$ for every $\hat{\varphi}_i \in [\underline{\varphi_i}^{(t)}, \overline{\varphi_i}^{(t)}]$.*

While Theorem 3.3 proves the termination of our approach, the runtime required until termination can be exponential in the number of features, due to the theoretical complexity of computing SHAP values exactly (Van den Broeck et al., 2022; Arenas et al., 2023). However, as we show next, our approach terminates in the first iteration for linear models. This result implies that our approach can terminate early even for non-linear models, particularly when the model can be decomposed into linear regions, as is the case for ReLU neural networks, which are inherently piecewise-linear.

**Proposition 3.5** (Linear Models). *If $[\underline{\Delta_i}, \overline{\Delta_i}]$ are computed using an LBP method and $f(\mathbf{x}) = \mathbf{w}^\mathsf{T}\mathbf{x} + \mathbf{b}$, it holds that $\underline{\varphi_i}^{(1)} = \overline{\varphi_i}^{(1)} = \varphi_i$.*

*Proof Sketch.* We first prove that the contribution $\Delta_i(S)$ is constant in $S$ for linear models. Since LBP methods have no approximation error for compositions of linear functions (Zhang et al., 2018)—such as the contribution of a linear model—LBP computes tight bounds equal to the constant contribution for every branch. This provides the exact SHAP value in the first iteration for a linear model.

An equivalent argument proves that our approach terminates immediately if, after sufficient splitting, $\Delta_i$ becomes linear in each branch, as illustrated in Figure 1. This can be the case, e.g., for piecewise-linear functions, such as ReLU-activated neural networks, when each branch lies within a single linear segment. Specifically, the runtime of VERISHAP is upper-bounded by $\mathcal{O}(B \cdot T_{BP})$, where $B$ denotes the number of explored branches and $T_{BP}$ is the cost of bound propagation per branch. The central question is how large $B$ becomes in practice: if branching requires only a few iterations to restrict the network to regions where the marginal contribution is linear, or sufficiently close to linear for the bound propagation to produce tight bounds, $B$ remains small and VERISHAP is efficient in practice.

Section 6 demonstrates that $B$ indeed remains comparatively small in practice, allowing our approach to outperform the existing state-of-the-art for computing exact SHAP values. Before moving on to Section 6, we first treat more practical aspects of implementing our approach.

## 4. Efficient SHAP Bounding

While Section 3 introduces our approach at a conceptual level, this section fills in details for practically bounding SHAP values of neural networks and introduces our concrete VERISHAP algorithm. Specifically, we discuss *(i)* leveraging bound propagation for computing bounds on $\Delta_i$; *(ii)* pruning branches to manage memory demands; *(iii)* batch-processing branches to leverage parallel hardware; *(iv)* computing sums of coalition weights $\Lambda_\mathcal{B}$ particularly efficiently; and *(v)* reusing branches for simultaneously bounding the SHAP values of all features. Section 5 dis-

**Algorithm 1** VERISHAP

---

**Input:** Value function $\mathbf{v} : \{0,1\}^n \to \mathbb{R}$, batch size $b$

$[\underline{\mathbf{m}}, \overline{\mathbf{m}}] \leftarrow [\mathbf{0}_n, \mathbf{1}_n]; \quad \Lambda \leftarrow 1$

$[\underline{\mathbf{v}}, \overline{\mathbf{v}}] \leftarrow \text{BOUNDPROPAGATION}(\mathbf{v}, [\underline{\mathbf{m}}, \overline{\mathbf{m}}])$

$[\underline{\boldsymbol{\varphi}}^{(1)}, \overline{\boldsymbol{\varphi}}^{(1)}] \leftarrow \text{ASSEMBLE}(\Lambda, \underline{\mathbf{v}}, \overline{\mathbf{v}})$ *// Proposition 4.3*

$\text{B}^{(1)} \leftarrow \{(\underline{\mathbf{m}}, \overline{\mathbf{m}}, \Lambda, \underline{\mathbf{v}}, \overline{\mathbf{v}})\}$

**for** $t \in \{2, \ldots, 2^n\}$ **do**

   $(\underline{\mathbf{M}}, \overline{\mathbf{M}}, \boldsymbol{\Lambda}, \underline{\mathbf{v}}, \overline{\mathbf{v}}), others \leftarrow \text{SELECT}(\text{B}^{(t-1)}, b)$

   $[\underline{\mathbf{M}}', \overline{\mathbf{M}}'] \leftarrow \text{SPLIT}(\underline{\mathbf{M}}, \overline{\mathbf{M}})$

   $[\underline{\mathbf{v}}', \overline{\mathbf{v}}'] \leftarrow \text{BOUNDPROPAGATION}(\mathbf{v}, [\underline{\mathbf{M}}', \overline{\mathbf{M}}'])$

   $\boldsymbol{\Lambda}' \leftarrow \text{REFINE}(\boldsymbol{\Lambda})$ *// Corollary 4.2*

   $[\underline{\boldsymbol{\varphi}}, \overline{\boldsymbol{\varphi}}] \leftarrow \text{ASSEMBLE}(\boldsymbol{\Lambda}, \underline{\mathbf{v}}, \overline{\mathbf{v}})$

   $[\underline{\boldsymbol{\varphi}}', \overline{\boldsymbol{\varphi}}'] \leftarrow \text{ASSEMBLE}(\boldsymbol{\Lambda}', \underline{\mathbf{v}}', \overline{\mathbf{v}}')$

   $\underline{\boldsymbol{\varphi}}^{(t)} \leftarrow \underline{\boldsymbol{\varphi}}^{(t-1)} - \underline{\boldsymbol{\varphi}} + \underline{\boldsymbol{\varphi}}'$

   $\overline{\boldsymbol{\varphi}}^{(t)} \leftarrow \overline{\boldsymbol{\varphi}}^{(t-1)} - \overline{\boldsymbol{\varphi}} + \overline{\boldsymbol{\varphi}}'$

   $\text{B}^{(t)} \leftarrow others \cup \text{PRUNE}((\underline{\mathbf{M}}', \overline{\mathbf{M}}', \boldsymbol{\Lambda}', \underline{\mathbf{v}}', \overline{\mathbf{v}}'))$

**end for**

---

cusses splitting strategies for partitioning and Appendix C further elaborates on selected aspects of VERISHAP.

Algorithm 1 summarises VERISHAP. In Algorithm 1, BOUNDPROPAGATION refers to a bound propagation technique as introduced in Section 2.2, such as CROWN. AS-SEMBLE and REFINE implement Proposition 4.3 and Corollary 4.2, respectively. SELECT and SPLIT are introduced in Section 5. All remaining sub-procedures and variables are described in the following paragraphs.

**Computing $\underline{\Delta}_{i\mathcal{B}}, \overline{\Delta}_{i\mathcal{B}}$.** We use bound propagation methods for neural networks as introduced in Section 2.2 to compute $\underline{\Delta}_{i\mathcal{B}} \leq \Delta_i(S) \leq \overline{\Delta}_{i\mathcal{B}}, \forall S \in \mathcal{B}$ where $\mathcal{B} \subseteq \mathcal{S}_i$ is a branch in the partition $\text{B}^{(t)}$ defined by the sets of included and excluded features $\mathcal{I}, \mathcal{E} \subseteq [n] \setminus \{i\}$. The challenge here is that bound propagation requires a function with real-valued vector arguments, while $\Delta_i$ has integer sets as arguments. To resolve this, we represent sets of features $S$ using *masks* $\mathbf{m} \in \{0,1\}^n$, such that $\mathbf{m}_j = 1 \Leftrightarrow j \in S, \forall j \in [n]$, and define $\boldsymbol{\Delta}_i : \{0,1\}^n \to \mathbb{R}$ as $\boldsymbol{\Delta}_i(\mathbf{m}) = \Delta_i(S)$. The masks allow us to identify branches $\mathcal{B}$ with the bounds $\underline{\mathbf{m}} \leq \mathbf{m} \leq \overline{\mathbf{m}}$, where $\underline{\mathbf{m}}, \overline{\mathbf{m}} \in \{0,1\}^n$, $\underline{\mathbf{m}}_j = 1 \Leftrightarrow j \in \mathcal{I}$, and $\overline{\mathbf{m}}_j = 0 \Leftrightarrow j \in \mathcal{E}$. By relaxing the Boolean masks to the continuous domain $[0,1]^n \subset \mathbb{R}^n$, we can perform bound propagation on $\boldsymbol{\Delta}_i$ using $[\underline{\mathbf{m}}, \overline{\mathbf{m}}]$ as input bounds.

**Example 4.1.** To better illustrate our use of masks, we show how to compute $\boldsymbol{\Delta}_i$ in practice. Let $\mathbf{m} \in \{0,1\}^n$ represent the set of features $S \in \mathcal{S}_i$. Now, $\boldsymbol{\Delta}_i(\mathbf{m}) = \mathbf{v}(\mathbf{m}^{+i}) - \mathbf{v}(\mathbf{m})$, where $\mathbf{m}_i^{+i} = 1$, $\mathbf{m}_j^{+i} = \mathbf{m}_j$ for $j \in [n] \setminus \{i\}$, and $\mathbf{v} : \{0,1\}^n \to \mathbb{R}$ is $\mathbf{v}(\mathbf{m}) = \mathbb{E}_{\mathbf{z} \sim \mathcal{D}}[f(\mathbf{m} * \mathbf{x} + (1 - \mathbf{m}) * \mathbf{z})]$ with $*$ denoting element-wise multiplication and $\mathcal{D}$ is the data distribution. Here $\mathbf{m}^{+i}$ corresponds to $S \cup \{i\}$

in Equation (2), and $\mathbf{m} * \mathbf{x} + (1 - \mathbf{m}) * \mathbf{z} = (\mathbf{x}_S; \mathbf{z}_{\bar{S}})$. Appendix C provides further details.

**Pruning Branches (PRUNE).** For high-dimensional feature spaces, VERISHAP may create a large number of branches leading to substantial memory demands. However, since branches $\mathcal{B}$ with $\underline{\Delta}_{i\mathcal{B}} = \overline{\Delta}_{i\mathcal{B}}$ do not have to be refined further, these branches can be *pruned*. If PB is the family of all pruned branches, it suffices to store $\sum_{\mathcal{B} \in \text{PB}} \Lambda_{\mathcal{B}} \underline{\Delta}_{i\mathcal{B}}$ instead of the individual branches in memory. In Algorithm 1, this value is stored implicitly in $[\underline{\boldsymbol{\varphi}}^{(t)}, \overline{\boldsymbol{\varphi}}^{(t)}]$.

**Batch Processing.** Bound propagation and branch and bound are particularly well-suited for modern massively-parallel hardware since they allow for processing many branches in parallel (Xu et al., 2021). To optimally utilise such hardware, we implement VERISHAP to split the branches in our partition in batches. In Algorithm 1, $[\underline{\mathbf{M}}, \overline{\mathbf{M}}], \boldsymbol{\Lambda}$ and $[\underline{\mathbf{v}}, \overline{\mathbf{v}}]$ represent batches of feature mask bounds $[\underline{\mathbf{m}}, \overline{\mathbf{m}}]$, $\Lambda_{\mathcal{B}}$ values, and value bounds $[\underline{v}, \overline{v}]$, respectively.

**Computing $\Lambda_{\mathcal{B}}$.** Proposition 3.2 provides an efficient way to compute $\Lambda_{\mathcal{B}}$. It also gives rise to a recursive formula for $\Lambda_{\mathcal{B}}$ that further accelerates the computation and lends itself well to batch-processing.

**Corollary 4.2** (Recursive Computation of $\Lambda_{\mathcal{B}}$). *Consider $\mathcal{B} \in \text{B}^{(t)}$ defined by $\mathcal{I}$ and $\mathcal{E}$ with $r = |\mathcal{I}|$ and $s = |\mathcal{I}| + |\mathcal{E}|$. For $\mathcal{B}'$ and $\mathcal{B}''$ being the branches derived from $\mathcal{B}$ by including, respectively, excluding some feature, we have*

$$\Lambda_{\mathcal{B}'} = \frac{r+1}{s+2}\Lambda_{\mathcal{B}} \qquad \Lambda_{\mathcal{B}''} = \frac{s+1-r}{s+2}\Lambda_{\mathcal{B}}.$$

*Proof Sketch.* Corollary 4.2 directly follows from Proposition 3.2 by applying two binomial identities.

**Simultaneously Bounding $\varphi_1, \ldots, \varphi_n$.** Section 3 describes computing bounds on $\varphi_i$ for a fixed $i \in [n]$. This procedure can be executed in parallel for several features, e.g., $i_1, i_2 \in [n]$. However, the branches computed for $i_1$ and $i_2$ overlap significantly: all branches that exclude both $i_1$ and $i_2$ are identical in the parallel runs. We can avoid this duplication of effort by partitioning the entire set $\mathcal{P}([n])$ instead of $\mathcal{S}_i$, and computing bounds on $\varphi_i$ for every $i \in [n]$ by selecting the branches that contain, respectively, exclude the feature $i$. In this case, we apply bound propagation to bound the *value function* $v(S)$ instead of the contribution $\Delta_i(S)$ for each branch. Let $t \in \mathbb{N}$, let $\text{B}^{(t)}$ be a partition of $\mathcal{P}([n])$, and, for $\mathcal{B} \in \text{B}^{(t)}$, let $\underline{v}_{\mathcal{B}} \leq v(S) \leq \overline{v}_{\mathcal{B}}, \forall S \in \mathcal{B}$ be the value bounds computed by bound propagation. Using that $\Delta_i(S) = v(S \cup \{i\}) - v(S)$ and carefully accounting for $\Lambda_{\mathcal{B}}$ following Corollary 4.2, we obtain the following result by following similar steps as for proving Theorem 3.1.

**Proposition 4.3.** *Let* $\mathsf{B}^{(t)}$ *be a partition of* $\mathcal{P}([n])$. *We have*

$$\underline{\varphi}_i^{(t)} := \sum_{\mathcal{B} \in \mathsf{B}_{+i}^{(t)}} \Lambda_{\mathcal{B}}^- \underline{v}_{\mathcal{B}} + \sum_{\mathcal{B} \in \mathsf{B}_{\pm i}^{(t)}} \Lambda_{\mathcal{B}} (\underline{v}_{\mathcal{B}} - \overline{v}_{\mathcal{B}}) - \sum_{\mathcal{B} \in \mathsf{B}_{-i}^{(t)}} \Lambda_{\mathcal{B}}^+ \overline{v}_{\mathcal{B}} \le \varphi_i$$

$$\overline{\varphi}_i^{(t)} := \sum_{\mathcal{B} \in \mathsf{B}_{+i}^{(t)}} \Lambda_{\mathcal{B}}^- \overline{v}_{\mathcal{B}} + \sum_{\mathcal{B} \in \mathsf{B}_{\pm i}^{(t)}} \Lambda_{\mathcal{B}} (\overline{v}_{\mathcal{B}} - \underline{v}_{\mathcal{B}}) - \sum_{\mathcal{B} \in \mathsf{B}_{-i}^{(t)}} \Lambda_{\mathcal{B}}^+ \underline{v}_{\mathcal{B}} \ge \varphi_i,$$

*where* $\mathsf{B}_{+i}^{(t)} \subseteq \mathsf{B}^{(t)}$ *contains all* $\mathcal{B} \in \mathsf{B}^{(t)}$ *with* $i \in \mathcal{I}$, $\mathsf{B}_{-i}^{(t)} \subseteq \mathsf{B}^{(t)}$ *contains all* $\mathcal{B} \in \mathsf{B}^{(t)}$ *with* $i \in \mathcal{E}$, $\mathsf{B}_{\pm i}^{(t)} \subseteq \mathsf{B}^{(t)}$ *contains the remaining branches*, $\Lambda_{\mathcal{B}}^- := \Lambda_{\mathcal{B}}(s + 1)/r$, *and* $\Lambda_{\mathcal{B}}^+ := \Lambda_{\mathcal{B}}(s + 1)/(s - r)$ *with* $r$ *and* $s$ *as in Proposition 3.2*.

## 5. Splitting Strategies

Refining partitions as described in Section 3 requires selecting *(i)* branches to split and *(ii)* features to split on. This section briefly introduces several strategies for performing these selections, which are discussed in detail in Appendix B and compared empirically in Appendix E.1.

**Selecting Branches to Split (SELECT).** The partitioning strategy in Section 3 requires selecting a branch $\mathcal{B}_k$ to split. Since we process branches in batches, we instead select a batch of $b \in \mathbb{N}$ branches to split simultaneously. We present two strategies for selecting branches: MAX-DIAM and MINDIAM select the $b$ branches where $d_{\mathcal{B}} := \Lambda_{\mathcal{B}}(\overline{\Delta}_{i_{\mathcal{B}}} - \underline{\Delta}_{i_{\mathcal{B}}})$ is the largest, respectively, the smallest.

**Selecting Features to Split on (SPLIT).** The branch-and-bound literature provides numerous strategies for selecting features to split, which can also be applied in VERISHAP. We present a baseline strategy, INORDER, which selects features in a predetermined order, and adapt strong branching (Applegate et al., 1995), smart branching (Bunel et al., 2020; Boetius et al., 2025), and smears (Kearfott, 1996; Wang et al., 2018a) to our setting. While strong branching and smart branching simulate splitting every available feature, smears leverages bounds of the contribution's gradient to determine which feature to split on. Concretely, for the branch $\mathcal{B}$, the smears strategy selects $j^* := \arg \max_j \overline{|\nabla \Delta_i|}_j$, where $\overline{|\nabla \Delta_i|} \ge |\nabla \Delta_i(S)|$ is computed using IBP. Appendix B provides further details.

## 6. Experiments

In this section, we investigate the practical scalability of VERISHAP, comparing it to state-of-the-art approaches for exactly computing and estimating SHAP values. Our theoretical results in Section 3 establish that VERISHAP can compute tight bounds on SHAP values, potentially without exploring all feature sets. In the following, we

*(i)* investigate the practical convergence and tightness of

VERISHAP's bounds,

*(ii)* compare VERISHAP to the state-of-the-art approach EXACTSHAP (Lundberg et al., 2025) for computing exact SHAP values,

*(iii)* investigate the relationship between VERISHAP and SHAP estimators,

*(iv)* compare SHAP estimators using the 'ground-truth' exact SHAP values computed by VERISHAP on larger search spaces than were previously feasible, and

*(v)* demonstrate the versatility of VERISHAP by applying it to different neural network architectures.

Appendix D provides details on the neural networks and datasets used in this section. Appendix E contains additional experiments, such as a comparison of the splitting strategies introduced in Section 5. In summary, we find that

*(i)* the bounds computed by VERISHAP provide helpful insights long before VERISHAP terminates;

*(ii)* VERISHAP scales to significantly larger search spaces than EXACTSHAP, both for computing exact SHAP values and, especially, for computing tight bounds;

*(iii)* VERISHAP complements SHAP estimators when high accuracy and precision are required;

*(iv)* VERISHAP provides novel insights into SHAP estimators by enabling evaluation on neural networks for larger search spaces; and

*(v)* VERISHAP can compute tight bounds for a variety of network architectures.

**Experimental Setup.** We implement VERISHAP in Python using JAX (Bradbury et al., 2025), leveraging a batched priority queue (Chen et al., 2021) for storing branches, and rely on the SHAP estimator implementations of Lundberg et al. (2025) and Witter et al. (2025). Following our ablation experiments in Appendix E.1, we use the CROWN-IBP (Zhang et al., 2020) LBP algorithm for computing bounds, MAXDIAM for branch selection, and SMEARS for selecting splits. We run our experiments on an L40S NVIDIA GPU with 48GB of GPU memory deployed on an Ubuntu 24.04 machine. Our code is available at `https://github.com/sen-uni-kn/verishap/`.

### 6.1. Insights into SHAP Values from Bounds

For studying the convergence and tightness of the VERISHAP bounds, we apply it to an MNIST (LeCun et al., 1998) convolutional neural network using the zero-baseline

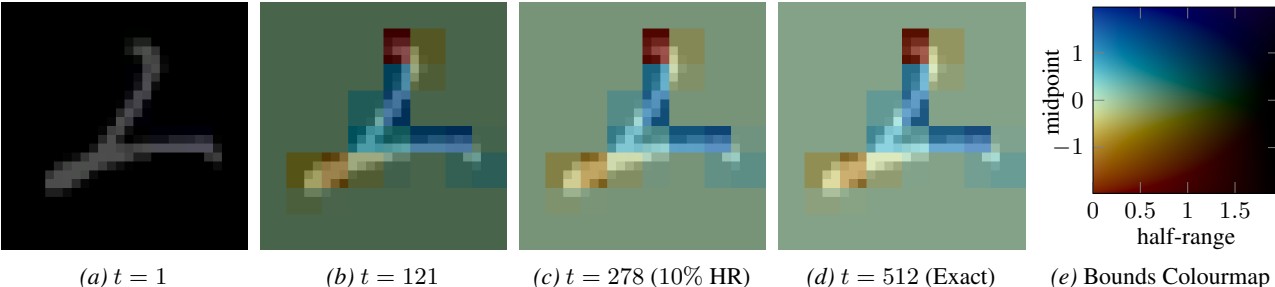

(a) $t = 1$      (b) $t = 121$      (c) $t = 278$ (10% HR)      (d) $t = 512$ (Exact)      (e) Bounds Colourmap

*Figure 2.* **MNIST VERISHAP Results.** SHAP value bounds for the class '4' score of an MNIST CNN on a test set image of a '2'. The SHAP values are computed for an $8 \times 8$ grid of superpixels. The SHAP bounds $\underline{\varphi_i}, \overline{\varphi_i}$ are visualised by their midpoints $(\overline{\varphi_i} + \underline{\varphi_i})/2$ and half-ranges (HR) $(\overline{\varphi_i} - \underline{\varphi_i})/2$ determining colour hue and lightness, respectively. **Lighter colour means tighter bounds in this figure.** In (c), the largest half-range is less than 10% of the network output for '4' ('10% HR') and in (d), the bounds provide the exact SHAP value. An animated version of this figure is available in the supplementary material.

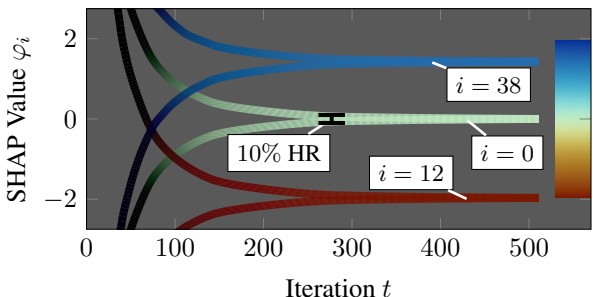

*Figure 3.* VERISHAP bounds for three features from Figure 2.

SHAP value function (Sundararajan & Najmi, 2020). Appendix E.5 provides further results on additional datasets and value functions. To manage the input dimensionality, we compute SHAP values for an $8 \times 8$ grid of evenly spaced superpixels. While the overall number of coalitions $2^{64} > 10^{19}$ is still unmanageably large, VERISHAP is able to compute the exact SHAP values within 34s. Furthermore, the bounds computed by VERISHAP prove insightful many iterations before the exact SHAP values are computed, as visualised in Figures 2 and 3. While the initial bounds are too loose to be insightful, after 120 iterations (25s), the bounds already reveal attribution patterns. In subsequent iterations, the bounds tighten further until the exact value is computed after 512 iterations (34s). We found that VERISHAP can compute the exact SHAP values in this setting because CROWN-IBP detects that the value function is constant across many coalitions. As a result of Proposition 3.5, this enables VERISHAP to compute exact SHAP values without enumerating all coalitions.

### 6.2. Comparison to EXACTSHAP

We now compare VERISHAP to the existing state-of-the-art algorithm EXACTSHAP for computing exact SHAP values. EXACTSHAP is an optimised algorithm for enumerating all coalitions implemented in the shap library (Lundberg et al.,

2025). We train neural networks on a variety of widely used tabular datasets from the UCI ML repository (Kelly et al.) and compute SHAP values for the first 10 test set samples for each network, evaluating the marginal value function using a background dataset of 100 samples.

Table 1 provides the median runtimes of EXACTSHAP and VERISHAP. Appendix E.6 provides additional statistics. While EXACTSHAP computes the exact SHAP values faster than VERISHAP for up to $|\mathcal{P}([n])| = 2^{20} \simeq 10^6$ coalitions, EXACTSHAP exhausts the GPU memory for larger search spaces. In contrast, VERISHAP is able to compute exact SHAP values for instances with up to $2^{25} \simeq 3 \cdot 10^7$ coalitions and provides tight bounds on SHAP values for up to $2^{60} \simeq 10^{18}$ coalitions. As Table 1 shows, the runtime of VERISHAP is not determined solely by the search space, but varies by dataset. Although we also encountered three datasets (HepatitisC, LungCancer, Online News) with fewer than 60 features for which VERISHAP was unable to compute tight bounds, VERISHAP provides tight SHAP bounds for the majority of datasets.

### 6.3. Comparison to KERNELSHAP and TREEMSR

Section 6.2 shows that VERISHAP outperforms the state-of-the-art approach for computing exact SHAP values. Since it is more common to statistically estimate SHAP values than to compute them exactly, we now compare VERISHAP to the popular KERNELSHAP (Lundberg & Lee, 2017) and the recent TREEMSR (Witter et al., 2025) estimators on the three highest-dimensional datasets from Table 1 for which we obtained exact SHAP values.

Figure 4 presents our results. The figure shows that both estimators cannot achieve the same accuracy as VERISHAP within the memory constraints of our hardware. While an estimator run requires significantly less time than VERISHAP, it does not provide an indication of the estimate variance. When repeating the estimation 100 times, as in

*Table 1.* **VERISHAP vs. EXACTSHAP.** We report the median runtime required by EXACTSHAP and VERISHAP for computing ('Exact') or tightly bounding ('10% HR', '1% HR') the marginal SHAP values on tabular datasets of dimension $n$, where '$p$% HR' denotes the half-range $(\overline{\varphi_i} - \underline{\varphi_i})/2$ of the SHAP bounds $\underline{\varphi_i}, \overline{\varphi_i}$ being at most $p$% of the network output to attribute, and '–' denotes exhausting the GPU memory or reaching the timeout of 600s.

| | | **EXACT** | **VERISHAP (Ours)** | | |
|---|---|---|---|---|---|
| $\|\mathcal{P}([n])\|$ | **Dataset** | **SHAP** | **10% HR** | **1% HR** | **Exact** |
| $2^{16}$ >$10^4$ | Obesity | 4s | 18s | 18s | 18s |
| $2^{20}$ >$10^6$ | German | 9s | 16s | 19s | 20s |
| $2^{22}$ >$10^6$ | Mushroom | – | 17s | 20s | 25s |
| $2^{23}$ >$10^6$ | Default | – | 127s | 132s | 133s |
| $2^{25}$ >$10^7$ | Auto | – | 81s | 213s | 316s |
| $2^{27}$ >$10^8$ | Steel | – | 37s | 322s | – |
| $2^{30}$ >$10^9$ | BreastCancer | – | 49s | 322s | – |
| $2^{38}$ >$10^{11}$ | Annealing | – | 269s | – | – |
| $2^{60}$ >$10^{18}$ | Sonar | – | 13s | – | – |

Figure 4, KERNELSHAP requires 346s, 123s, and 351s on Mushroom, Default, and Auto, respectively, at the largest feasible sample sizes. Similarly, TREEMSR requires 895s, 132s, and 341s on Mushroom, Default, and Auto, respectively. For both estimators, this exceeds the overall runtime of VERISHAP on Mushroom and Auto. Overall, VERISHAP complements statistical estimators when high accuracy and precision are required. Appendix E.7 contains equivalent results for additional statistical SHAP estimators.

### 6.4. Evaluating SHAP Estimators Using VERISHAP

We evaluate statistical SHAP estimators using the exact SHAP values computed by VERISHAP on datasets where obtaining exact SHAP values was previously infeasible. Following Witter et al. (2025), we compute the value function using the dataset mean as a singleton background dataset and report mean squared errors (MSE) in this experiment. As Figure 5 shows, LEVERAGESHAP (Musco & Witter, 2025) outperforms KERNELSHAP, confirming a trend emerging on smaller search spaces (Witter et al., 2025; Musco & Witter, 2025). TREEMSR shows volatile performance across datasets when applied to neural networks, a nuanced insight that is not directly apparent from the results of Witter et al. (2025) for explaining tree-based models with large search spaces. This exemplifies the potential of the 'ground-truth' exact SHAP values computed by VERISHAP on larger search spaces to provide insights that are not apparent in tree-based ML models and neural networks on lower-dimensional datasets.

### 6.5. Activation Functions and Network Architectures

To demonstrate the versatility of VERISHAP, we apply VERISHAP to Mushroom networks with different activ-

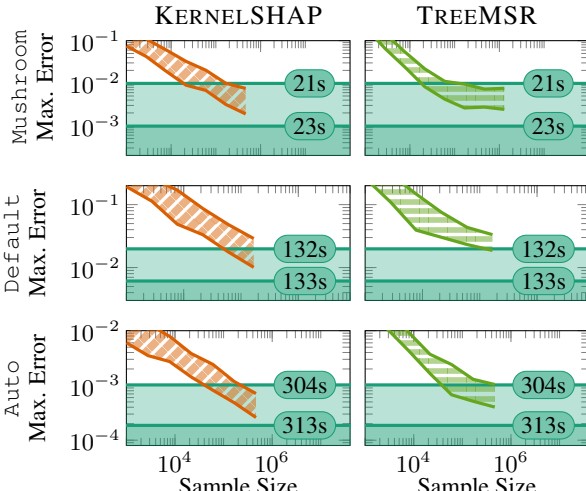

*Figure 4.* **VERISHAP** ▪ **vs. KERNELSHAP** ▨ **& TREEMSR** ▨ We run KERNELSHAP and TREEMSR with increasing sample sizes until each exhausts the available GPU memory, repeating each run 100 times. For each sample size, we plot the range across repetitions of the *largest error* between the estimated SHAP value and the true SHAP value for any feature. For VERISHAP, we mark the half-ranges of the SHAP bounds at different ⬭runtimes⬭.

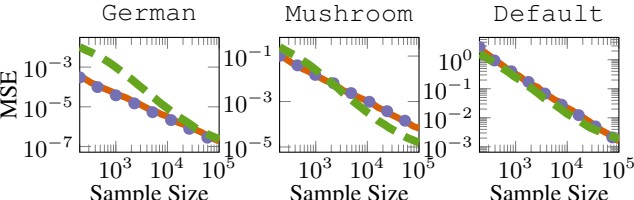

*Figure 5.* **KERNELSHAP** ━ **vs. LEVERAGESHAP** •••• **vs. TREEMSR** ╍. We report the mean squared error (MSE) of the SHAP estimators across 100 runs on three representative datasets.

ation functions and architectures. Concretely, we apply VERISHAP to fully-connected networks with ReLU, tanh, and Swish (Hendrycks & Gimpel, 2016) as activation function, as well as a RestNet (He et al., 2016) with fully-connected layers. Table 2 provides the runtime of VERISHAP for the different networks. While the runtime of VERISHAP increases for the non-piecewise-linear tanh and Swish-activated neural networks, as well as the larger ResNet architecture, VERISHAP is able to compute exact SHAP values for all networks. Appendix E.3 provides a complementary experiment comparing the scalability of VERISHAP in the network size.

## 7. Related Work

**Exact SHAP.** To obtain Shapley values while preserving their full game-theoretic guarantees, one must compute them exactly. While exact computation is feasible, e.g., for tree-based models (Yu et al., 2022) and additive models (Enouen & Liu, 2025; Bordt & von Luxburg, 2023), com-

*Table 2.* **VERISHAP for Different Activation Functions and Network Architectures.** In this table, 'ReLU', 'tanh', and 'Swish' represent fully-connected networks with the respective activation functions, and 'ResNet' represents a ReLU-activated fully-connected network with residual connections, all trained on `Mushroom`. The column headings are as for Table 1. The timeout for this experiment is $900s$.

| | VERISHAP (Ours) Runtime | | | |
|---|---|---|---|---|
| **Network** | 10% HR | 1% HR | 0.1% HR | Exact |
| ReLU | 17s | 20s | 24s | 25s |
| tanh | 19s | 37s | 61s | 670s |
| Swish | 18s | 35s | 56s | 70s |
| ResNet | 72s | 135s | 188s | 199s |

puting exact Shapley values is #P-hard for general neural networks (Van den Broeck et al., 2022; Arenas et al., 2023; Marzouk et al., 2025b). Unlike existing approaches that rely on restricted neural network architectures to enable exact SHAP computation (Marzouk et al., 2025a; Chen et al., 2023; Heidari et al., 2025; Muschalik et al., 2025), our algorithm is applicable to general neural networks.

**SHAP Value Estimation Methods.** Due to the computational complexity of computing exact SHAP values, a multitude of approaches exist for estimating SHAP values, including KERNELSHAP (Lundberg & Lee, 2017; Covert & Lee, 2021), TREEMSR and LINEARMSR (Witter et al., 2025), LEVERAGESHAP (Musco & Witter, 2025), FASTSHAP (Jethani et al., 2022), DEEPSHAP (Chen et al., 2020), as well as Monte Carlo-based approaches (Štrumbelj & Kononenko, 2014; Mitchell et al., 2022a), uncertainty-based estimates (Ancona et al., 2019; Watson et al., 2023), and further model-specific methods for kernel models (Chau et al., 2022; 2023). Our work addresses the substantially more challenging task of computing *exact* SHAP values, and has the potential to serve as a 'ground-truth' framework for evaluating SHAP estimators.

**Neural Network Verification & Formal Explainability.** Our approach is based on the recent rapid progress in the field of neural network verification (Brix et al., 2024; Wang et al., 2021; Zhou et al., 2025; 2024; Singh et al., 2019; Müller et al., 2021; Chiu et al., 2025; Ferrari et al., 2022; Boetius et al., 2025; Wu et al., 2024). Recently, related techniques have also been used in the context of formal explainability (Marques-Silva & Ignatiev, 2022; Ignatiev et al., 2019; Darwiche & Hirth, 2020; Bassan & Katz, 2023; Audemard et al., 2022), a subfield that seeks explanations with formal guarantees. However, in this context (see, e.g., Bassan et al., 2025; Wu et al., 2023; Izza et al., 2024; Malfa et al., 2021; Labbaf et al., 2025; Hadad et al., 2026; Soria et al., 2026), neural network verifiers are used for explanation types that relate directly to adversarial robustness,

which simplifies the incorporation of neural network verifiers. Our work is the first to apply ideas from neural network verification to computing exact SHAP values.

## 8. Limitations and Future Work

Since computing exact SHAP values is #P-hard, all algorithms for computing them, including VERISHAP, face substantial computational challenges. As VERISHAP relies on neural network verification, which has made rapid progress in recent years (Brix et al., 2024; Wang et al., 2021; Zhou et al., 2025; 2024; Chiu et al., 2025), and has demonstrated remarkable success on other NP-hard (Katz et al., 2017) and #P-hard problems (Boetius et al., 2025), its scalability will improve as neural network verification continues to advance. Importantly, VERISHAP already scales to substantially larger search spaces than existing exact SHAP approaches. While VERISHAP can still be slow for some networks, users can diagnose such cases early from the initial SHAP bounds, as demonstrated in Appendix E.9. We also acknowledge existing critiques of SHAP (Huang & Marques-Silva, 2024; Kumar et al., 2020; Slack et al., 2020; Fryer et al., 2021; Biradar et al., 2024); our goal is not to defend its axiomatic foundations, but to provide a way to compute provably exact values for a widely used explanation method.

Future work may extend our algorithm to higher-order Shapley interactions (Fumagalli et al., 2023; 2024; Sundararajan & Najmi, 2020; Kolpaczki et al., 2024), exact SHAP under alternative value functions (Mohammadi et al., 2025; Létoffé et al., 2025) and input distributions (Ghalebikesabi et al., 2021), as well as other feature attribution indices (Baniecki et al., 2025; Barceló et al., 2025).

## 9. Conclusion

We present VERISHAP, the first algorithm that leverages recent advances in neural network verification to compute *exact* SHAP values of neural networks. Our approach scales to search spaces orders of magnitude larger than those supported by existing methods, marking an important first step towards exact SHAP computation for neural networks. Moreover, our method flexibly provides provably exact bounds on SHAP values with arbitrary precision, enabling more scalable early termination while still yielding meaningful explanatory insights. Finally, our verification-based approach produces reliable 'ground-truth' explanations for evaluating SHAP estimators for neural networks on larger search spaces. Hence, our work is an important step forward for producing and evaluating trustworthy explanations for neural networks through provable, verification-based guarantees.

## Acknowledgements

The work of Leue and Boetius was partially funded through the DFG research grant LE 1342/4 'SCADNet - Structural Causal Analysis of Deep Neural Networks'. The work of Katz and Bassan was partially funded by the European Union (ERC, VeriDeL, 101112713). Views and opinions expressed are however those of the author(s) only and do not necessarily reflect those of the European Union or the European Research Council Executive Agency. Neither the European Union nor the granting authority can be held responsible for them. This research was additionally supported by a grant from the Israeli Science Foundation (grant number 558/24).

## Impact Statement

Since our work advances both the practical and theoretical aspects of AI explainability, it shares broader implications common to the field, such as vulnerability to adversarial manipulation, privacy risks, and potential bias. However, by computing *exact* SHAP values and certified bounds with provable guarantees, we aim to make these explanations more trustworthy and to provide a rigorous foundation for evaluating SHAP estimators in more realistic, higher-dimensional settings.

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

/forum?id=HuSSR12Yot.

# Appendix

## A. Proofs

We restate all theoretical results for easier reference in this section.

### A.1. Sums of Coalition Weights

**Proposition 3.2** (Restated)**.** *Consider $\mathcal{B} \in \mathsf{B}^{(t)}$ defined by $\mathcal{I}$ and $\mathcal{E}$. Letting $r := |\mathcal{I}|$ and $s := |\mathcal{I}| + |\mathcal{E}|$, we have*

$$\Lambda_{\mathcal{B}} = \sum_{S \in \mathcal{B}} \lambda(|S|) = \frac{1}{(s+1)\binom{s}{r}}. \tag{3}$$

*Proof.* By plugging the value of $\lambda$ into Equation (3), Proposition 3.2 becomes equivalent to showing that

$$\frac{1}{n}\sum_{k=0}^{n-1-s}\binom{n-1-s}{k}\binom{n-1}{k+r}^{-1} = \frac{1}{(s+1)\binom{s}{r}}. \tag{4}$$

By the definition of the binomial coefficient,

$$\binom{n-1}{k+r}^{-1} = \frac{(k+r)!(n-1-k-r)!}{(n-1)!}.$$

We also recall the following identity for the Beta function

$$\beta(a,b) = \int_0^1 t^{a-1}(1-t)^{b-1}dt = \frac{(a-1)!(b-1)!}{(a+b-1)!}, \tag{5}$$

as well as the classical truncated binomial theorem, which for $x \in \mathbb{R}$ takes the form

$$\sum_{k=0}^{K}\binom{K}{k}x^k = (1+x)^K. \tag{6}$$

Setting $a := k+r+1$ and $b := n-k-r$ in Equation (5) yields

$$\beta(k+r+1, n-k-r) = \frac{(k+r)!(n-k-r-1)!}{n!}$$

$$\Longleftrightarrow \qquad n \cdot \beta(k+r+1, n-k-r) = \frac{(k+r)!(n-k-r-1)!}{(n-1)!} = \binom{n-1}{k+r}^{-1}.$$

Hence, we obtain

$$\binom{n-1}{k+r}^{-1} = n\int_0^1 t^{k+r}(1-t)^{n-k-r-1}dt$$

$$\Longleftrightarrow \qquad \frac{1}{n}\cdot\binom{n-1}{k+r}^{-1} = \int_0^1 t^{k+r}(1-t)^{n-k-r-1}dt$$

$$\Longleftrightarrow \qquad \frac{1}{n}\cdot\binom{n-1}{k+r}^{-1}\cdot\binom{n-1-s}{k} = \binom{n-1-s}{k}\cdot\int_0^1 t^{k+r}(1-t)^{n-k-r-1}dt.$$

Substituting this result into the left-hand side of Equation (4) yields

$$\frac{1}{n}\sum_{k=0}^{n-1-s}\binom{n-1-s}{k}\binom{n-1}{k+r}^{-1} = \sum_{k=0}^{n-1-s}\binom{n-1-s}{k}\cdot\int_0^1 t^{k+r}(1-t)^{n-k-r-1}dt$$

$$= \sum_{k=0}^{n-1-s}\binom{n-1-s}{k}\cdot\int_0^1 t^r(1-t)^{n-r-1}\left(\frac{t}{1-t}\right)^k dt. \tag{7}$$

Setting $x := \frac{t}{1-t}$ and $K := n-1-s$ in Equation (6), we get

$$\sum_{k=0}^{n-1-s}\binom{n-1-s}{k}\left(\frac{t}{1-t}\right)^k = \left(\frac{1}{1-t}\right)^{n-1-s}.$$

Inserting this expression into Equation (7), we obtain

$$\frac{1}{n} \sum_{k=0}^{n-1-s} \binom{n-1-s}{k} \binom{n-1}{k+r}^{-1} = \sum_{k=0}^{n-1-s} \binom{n-1-s}{k} \cdot \int_0^1 t^r (1-t)^{n-r-1} \left(\frac{t}{1-t}\right)^k dt$$

$$= \int_0^1 t^r (1-t)^{n-r-1} \left(\frac{1}{1-t}\right)^{n-1-s} dt$$

$$= \int_0^1 t^r (1-t)^{s-r} dt.$$

Using Equation (5), we obtain

$$\int_0^1 t^r (1-t)^{s-r} dt = \beta(r+1, s-r+1) = \frac{r!(s-r)!}{(s+1)!} = \frac{r!(s-r)!}{s!(s+1)} = \frac{1}{\binom{s}{r}(s+1)}.$$

Therefore,

$$\Lambda_{\mathcal{B}} = \frac{1}{n} \sum_{k=0}^{n-1-s} \binom{n-1-s}{k} \binom{n-1}{k+r}^{-1} = \int_0^1 t^r (1-t)^{s-r} dt = \frac{1}{\binom{s}{r}(s+1)},$$

which completes the proof.

$\square$

**Corollary 4.2** (Restated). *Consider $\mathcal{B} \in \mathsf{B}^{(t)}$ defined by $\mathcal{I}$ and $\mathcal{E}$ with $r = |\mathcal{I}|$ and $s = |\mathcal{I}| + |\mathcal{E}|$. For $\mathcal{B}'$ and $\mathcal{B}''$ being the branches derived from $\mathcal{B}$ by including, respectively, excluding some feature, we have*

$$\Lambda_{\mathcal{B}'} = \frac{r+1}{s+2} \Lambda_{\mathcal{B}} \qquad \Lambda_{\mathcal{B}''} = \frac{s+1-r}{s+2} \Lambda_{\mathcal{B}}.$$

*Proof.* Let $\mathcal{B}, \mathcal{B}', \mathcal{B}'', r$, and $s$ be as in Corollary 4.2. By Proposition 3.2, we have

$$\Lambda_{\mathcal{B}'} = \frac{1}{(s+2)\binom{s+1}{r+1}} = \frac{r+1}{s+2} \frac{1}{(s+1)\binom{s}{r}}$$

$$= \frac{r+1}{s+2} \Lambda_{\mathcal{B}}$$

$$\Lambda_{\mathcal{B}''} = \frac{1}{(s+2)\binom{s+1}{r}} = \frac{s+1-r}{s+2} \frac{1}{(s+1)\binom{s}{r}}$$

$$= \frac{s+1-r}{s+2} \Lambda_{\mathcal{B}}.$$

$\square$

## A.2. SHAP Bounds

**Theorem 3.1** (Restated). *Let $\mathsf{B}^{(t)}$ be a partition of $\mathcal{S}_i$. It holds that*

$$\underline{\varphi_i}^{(t)} := \sum_{\mathcal{B} \in \mathsf{B}^{(t)}} \Lambda_{\mathcal{B}} \underline{\Delta}_{i\mathcal{B}} \leq \varphi_i \leq \sum_{\mathcal{B} \in \mathsf{B}^{(t)}} \Lambda_{\mathcal{B}} \overline{\Delta}_{i\mathcal{B}} =: \overline{\varphi_i}^{(t)}$$

*Proof.* By definition, we have $\Lambda_{\mathcal{B}} = \sum_{S \in \mathcal{B}} \lambda(|S|)$. It directly follows that

$$\underline{\varphi_i}^{(t)} = \sum_{\mathcal{B} \in \mathsf{B}^{(t)}} \sum_{S \in \mathcal{B}} \lambda(|S|) \underline{\Delta}_{i\mathcal{B}}$$

$$\overline{\varphi_i}^{(t)} = \sum_{\mathcal{B} \in \mathsf{B}^{(t)}} \sum_{S \in \mathcal{B}} \lambda(|S|) \overline{\Delta}_{i\mathcal{B}}.$$

Since $[\underline{\Delta}_{i_{\mathcal{B}}}, \overline{\Delta}_{i_{\mathcal{B}}}]$ are computed using bound propagation, it holds that $\underline{\Delta}_{i_{\mathcal{B}}} \leq \Delta_i(S) \leq \overline{\Delta}_{i_{\mathcal{B}}}, \forall S \in \mathcal{B}$. Hence, since $\lambda(k) > 0, \forall k \in \{0, \dots, n-1\}$, we obtain our final result by applying Equation (1)

$$\underline{\varphi_i}^{(t)} = \sum_{S \in \mathcal{S}_i} \lambda(|S|)\underline{\Delta}_{i_{\mathcal{B}}} \leq \varphi_i = \sum_{S \in \mathcal{S}_i} \lambda(|S|)\Delta_i(S) \leq \sum_{S \in \mathcal{S}_i} \lambda(|S|)\overline{\Delta}_{i_{\mathcal{B}}} = \overline{\varphi_i}^{(t)}.$$

$\square$

**Proposition 4.3** (Restated)**.** *Let* $\mathsf{B}^{(t)}$ *be a partition of* $\mathcal{P}([n])$. *We have*

$$\underline{\varphi}_i^{(t)} := \sum_{\mathcal{B} \in \mathsf{B}_{+i}^{(t)}} \Lambda_{\mathcal{B}}^- \underline{v}_{\mathcal{B}} + \sum_{\mathcal{B} \in \mathsf{B}_{\pm i}^{(t)}} \Lambda_{\mathcal{B}}(\underline{v}_{\mathcal{B}} - \overline{v}_{\mathcal{B}}) - \sum_{\mathcal{B} \in \mathsf{B}_{-i}^{(t)}} \Lambda_{\mathcal{B}}^+ \overline{v}_{\mathcal{B}} \leq \varphi_i$$

$$\overline{\varphi}_i^{(t)} := \sum_{\mathcal{B} \in \mathsf{B}_{+i}^{(t)}} \Lambda_{\mathcal{B}}^- \overline{v}_{\mathcal{B}} + \sum_{\mathcal{B} \in \mathsf{B}_{\pm i}^{(t)}} \Lambda_{\mathcal{B}}(\overline{v}_{\mathcal{B}} - \underline{v}_{\mathcal{B}}) - \sum_{\mathcal{B} \in \mathsf{B}_{-i}^{(t)}} \Lambda_{\mathcal{B}}^+ \underline{v}_{\mathcal{B}} \geq \varphi_i,$$

*where* $\mathsf{B}_{+i}^{(t)} \subseteq \mathsf{B}^{(t)}$ *contains all* $\mathcal{B} \in \mathsf{B}^{(t)}$ *with* $i \in \mathcal{I}$, $\mathsf{B}_{-i}^{(t)} \subseteq \mathsf{B}^{(t)}$ *contains all* $\mathcal{B} \in \mathsf{B}^{(t)}$ *with* $i \in \mathcal{E}$, $\mathsf{B}_{\pm i}^{(t)} \subseteq \mathsf{B}^{(t)}$ *contains the remaining branches*, $\Lambda_{\mathcal{B}}^- := \Lambda_{\mathcal{B}}(s+1)/r$, *and* $\Lambda_{\mathcal{B}}^+ := \Lambda_{\mathcal{B}}(s+1)/(s-r)$ *with* $r$ *and* $s$ *as in Proposition 3.2.*

*Proof.* We first recall Equation (1) and rewrite it as

$$\varphi_i = \sum_{S \in \mathcal{S}_i} \lambda(|S|)\Delta_i(S) = \sum_{S \in \mathcal{S}_i} \lambda(|S|)v(S \cup \{i\}) - \sum_{S \in \mathcal{S}_i} \lambda(|S|)v(S) = \underbrace{\sum_{S \in \mathcal{S}_i^+} \lambda(|S|-1)v(S)}_{(*)} - \underbrace{\sum_{S \in \mathcal{S}_i} \lambda(|S|)v(S)}_{(\ddagger)}, \quad (8)$$

where $\mathcal{S}_i^+ = \{S \cup \{i\} \mid S \in \mathcal{S}_i\}$ is the set of all feature sets, or *coalitions*, containing $i$. We now derive bounds on $(*)$ and $(\ddagger)$.

*Bounds on* $(*)$. Since $\mathsf{B}^{(t)}$ is a partition of $\mathcal{P}([n]) \supset \mathcal{S}_i^+$, we can partition $\mathcal{S}_i^+$ along $\mathsf{B}^{(t)}$. Since $\mathcal{S}_i^+$ is the set of all coalitions containing $i$, we have $\mathcal{S}_i^+ \cap \mathcal{B} \neq \emptyset$ for all $\mathcal{B} \in \mathsf{B}_{+i}^{(t)} \cup \mathsf{B}_{\pm i}^{(t)}$ and $\mathcal{S}_i^+ \cap \mathcal{B} = \emptyset$ for all $\mathcal{B} \in \mathsf{B}_{-i}^{(t)}$. In particular, by definition, the branches in $\mathsf{B}_{+i}^{(t)}$ *only* contain coalitions containing $i$ and *no* branch in $\mathsf{B}_{-i}^{(t)}$ contains coalitions containing $i$, while the branches in $\mathsf{B}_{\pm i}^{(t)}$ contain both coalitions with and without $i$. This allows us to partition $\mathcal{S}_i^+$ as follows

$$\sum_{S \in \mathcal{S}_i^+} \lambda(|S|-1)v(S) = \sum_{\mathcal{B} \in \mathsf{B}_{+i}^{(t)}} \sum_{S \in \mathcal{B}} \lambda(|S|-1)v(S) + \sum_{\mathcal{B} \in \mathsf{B}_{\pm i}^{(t)}} \sum_{S \in \mathcal{B} \cap \mathcal{S}_i^+} \lambda(|S|-1)v(S)$$

$$\geq \sum_{\mathcal{B} \in \mathsf{B}_{+i}^{(t)}} \underline{v}_{\mathcal{B}} \sum_{S \in \mathcal{B}} \lambda(|S|-1) + \sum_{\mathcal{B} \in \mathsf{B}_{\pm i}^{(t)}} \underline{v}_{\mathcal{B}} \sum_{S \in \mathcal{B} \cap \mathcal{S}_i^+} \lambda(|S|-1),$$

where the inequality holds since $\underline{v}_{\mathcal{B}} \leq v(S) \leq \overline{v}_{\mathcal{B}}, \forall S \in \mathcal{B}$, since $[\underline{v}_{\mathcal{B}}, \overline{v}_{\mathcal{B}}]$ is computed using bound propagation. We now derive a closed-form expression for the terms $\sum_S \lambda(|S|-1)$ in terms of $\Lambda_{\mathcal{B}}$ for $\mathcal{B} \in \mathsf{B}_{+i}^{(t)}$ and $\mathcal{B} \in \mathsf{B}_{\pm i}^{(t)}$. The main difficulty here is that, when $\mathsf{B}^{(t)}$ partitions $\mathcal{P}([n])$ instead of $\mathcal{S}_i$ and $\Lambda_{\mathcal{B}}$ is computed in terms of the number of included and excluded features following Proposition 3.2, $\Lambda_{\mathcal{B}}$ is effectively computed as if there were $n+1$ features instead of $n$. We follow through this reasoning for $\mathsf{B}_{+i}^{(t)}$ and $\mathsf{B}_{\pm i}^{(t)}$ separately.

- We first consider $\mathcal{B} \in \mathsf{B}_{+i}^{(t)}$ defined by $\mathcal{I} \ni i$ and $\mathcal{E} \not\ni i$. Consider an equivalent branch $\mathcal{B}^- \subseteq \mathcal{S}_i$ from a partitioning of $\mathcal{S}_i$ defined by $\mathcal{E}^- = \mathcal{E}$ and $\mathcal{I}^- = \mathcal{I} \setminus \{i\}$, since $i$ cannot be included in a branch when partitioning $\mathcal{S}_i$. It holds that $|\mathcal{B}| = |\mathcal{B}^-|$ and, since the coalitions in $\mathcal{B}^-$ contain exactly one element less than the corresponding coalitions in $\mathcal{B}$, we have $\sum_{S \in \mathcal{B}^-} \lambda(|S|) = \sum_{S \in \mathcal{B}} \lambda(|S|-1)$. Let $\Lambda_{\mathcal{B}}^- = \sum_{S \in \mathcal{B}^-} \lambda(|S|)$. By Proposition 3.2, we can compare $\Lambda_{\mathcal{B}}$ and $\Lambda_{\mathcal{B}}^-$ in terms of the number of included and excluded features. Let $r = |\mathcal{I}|$ and $s = |\mathcal{I}| + |\mathcal{E}|$. Since $|\mathcal{I}^-| = r - 1$ and $|\mathcal{I}^-| + |\mathcal{E}^-| = s - 1$, by Corollary 4.2, we have

$$\Lambda_{\mathcal{B}} = \frac{|\mathcal{I}^-|+1}{|\mathcal{I}^-|+|\mathcal{E}^-|+2}\Lambda_{\mathcal{B}}^- \quad \Longleftrightarrow \quad \Lambda_{\mathcal{B}} = \frac{r}{s+1}\Lambda_{\mathcal{B}}^- \quad \Longleftrightarrow \quad \frac{s+1}{r}\Lambda_{\mathcal{B}} = \Lambda_{\mathcal{B}}^-.$$

- Now, we consider $\mathcal{B} \in \mathsf{B}_{\pm i}^{(t)}$ defined by $\mathcal{I} \not\ni i$ and $\mathcal{E} \not\ni i$. There is a branch $\hat{\mathcal{B}} \subseteq \mathcal{S}_i$ from a partitioning of $\mathcal{S}_i$ defined by $\hat{\mathcal{I}} = \mathcal{I}$ and $\hat{\mathcal{E}} = \mathcal{E}$ that satisfies $\hat{\mathcal{B}} = \mathcal{B} \cap \mathcal{S}_i$. Since $|\hat{\mathcal{B}}| = |\mathcal{B} \cap \mathcal{S}_i| = |\mathcal{B} \cap \mathcal{S}_i^+|$, we have $\Lambda_{\hat{\mathcal{B}}} = \sum_{S \in \hat{\mathcal{B}}} \lambda(|S|) = \sum_{S \in \mathcal{B} \cap \mathcal{S}_i^+} \lambda(|S| - 1)$. By Proposition 3.2, we have $\Lambda_{\hat{\mathcal{B}}} = \Lambda_{\mathcal{B}}$.

Overall, we obtain

$$\sum_{S \in \mathcal{S}_i^+} \lambda(|S| - 1)v(S) \geq \sum_{\mathcal{B} \in \mathsf{B}_{+i}^{(t)}} \underline{v}_{\mathcal{B}} \sum_{S \in \mathcal{B}} \lambda(|S| - 1) + \sum_{\mathcal{B} \in \mathsf{B}_{\pm i}^{(t)}} \underline{v}_{\mathcal{B}} \sum_{S \in \mathcal{B} \cap \mathcal{S}_i^+} \lambda(|S| - 1) = \sum_{\mathcal{B} \in \mathsf{B}_{+i}^{(t)}} \Lambda_{\mathcal{B}}^- \underline{v}_{\mathcal{B}} + \sum_{\mathcal{B} \in \mathsf{B}_{\pm i}^{(t)}} \Lambda_{\mathcal{B}} \underline{v}_{\mathcal{B}}.$$

From an equivalent chain of arguments, we also obtain the upper bound

$$\sum_{S \in \mathcal{S}_i^+} \lambda(|S| - 1)v(S) \leq \sum_{\mathcal{B} \in \mathsf{B}_{+i}^{(t)}} \Lambda_{\mathcal{B}}^- \overline{v}_{\mathcal{B}} + \sum_{\mathcal{B} \in \mathsf{B}_{\pm i}^{(t)}} \Lambda_{\mathcal{B}} \overline{v}_{\mathcal{B}}.$$

*Bounds on* (‡). We follow similar steps as for (∗). As for $\mathcal{S}_i^+$, we can similarly partition $\mathcal{S}_i$ along $\mathsf{B}^{(t)}$. Since $\mathcal{S}_i = \mathcal{P}([n] \setminus \{i\})$, we have $\mathcal{S}_i \cap \mathcal{B} \neq \emptyset$ for all $\mathcal{B} \in \mathsf{B}_{-i}^{(t)} \cup \mathsf{B}_{\pm i}^{(t)}$ and $\mathcal{S}_i \cap \mathcal{B} = \emptyset$ for all $\mathcal{B} \in \mathsf{B}_{+i}^{(t)}$. Since $v(S) \geq \underline{v}_{\mathcal{B}}, \forall S \in \mathcal{B}$, we obtain

$$\sum_{S \in \mathcal{S}_i} \lambda(|S|)v(S) \geq \sum_{\mathcal{B} \in \mathsf{B}_{-i}^{(t)}} \underline{v}_{\mathcal{B}} \sum_{S \in \mathcal{B}} \lambda(|S|) + \sum_{\mathcal{B} \in \mathsf{B}_{\pm i}^{(t)}} \underline{v}_{\mathcal{B}} \sum_{S \in \mathcal{B} \cap \mathcal{S}_i} \lambda(|S|).$$

As for (∗), we express the terms $\sum_S \lambda(|S|)$ in terms of $\Lambda_{\mathcal{B}}$, while differentiating $\mathcal{B} \in \mathsf{B}_{-i}^{(t)}$ and $\mathcal{B} \in \mathsf{B}_{\pm i}^{(t)}$.

- Let $\mathcal{B} \in \mathsf{B}_{-i}^{(t)}$ be defined by $\mathcal{I} \not\ni i$ and $\mathcal{E} \ni i$. Consider a branch $\mathcal{B}^+$ containing the same coalitions as $\mathcal{B}$ but from a partitioning of $\mathcal{S}_i$. Being from a partitioning of $\mathcal{S}_i$, $\mathcal{B}^+$ is defined by $\mathcal{I}^+ = \mathcal{I}$ and $\mathcal{E}^+ = \mathcal{E} \setminus \{i\}$, since $i$ cannot be excluded in a partitioning of $\mathcal{S}_i$, as it is always excluded. Since $\mathcal{B}^+$ contains the same elements as $\mathcal{B}$, we trivially have $\sum_{S \in \mathcal{B}^+} \lambda(|S|) = \sum_{S \in \mathcal{B}} \lambda(|S|)$. Let $\Lambda_{\mathcal{B}}^+ = \sum_{S \in \mathcal{B}^+} \lambda(|S|)$, $r = |\mathcal{I}|$, and $s = |\mathcal{I}| + |\mathcal{E}|$. We have $|\mathcal{I}^+| = r$ and $|\mathcal{I}^+| + |\mathcal{E}^+| = s - 1$. By applying Proposition 3.2 and Corollary 4.2, we obtain

$$\Lambda_{\mathcal{B}} = \frac{|\mathcal{I}^+| + |\mathcal{E}^+| + 1 - |\mathcal{I}^+|}{|\mathcal{I}^+| + |\mathcal{E}^+| + 2} \Lambda_{\mathcal{B}}^+ \qquad \Longleftrightarrow \qquad \Lambda_{\mathcal{B}} = \frac{s - r}{s + 1} \Lambda_{\mathcal{B}}^+ \qquad \Longleftrightarrow \qquad \frac{s + 1}{s - r} \Lambda_{\mathcal{B}} = \Lambda_{\mathcal{B}}^+.$$

- Let $\mathcal{B} \in \mathsf{B}_{\pm i}^{(t)}$ be defined by $\mathcal{I} \not\ni i$ and $\mathcal{E} \not\ni i$. As for (∗), there is a branch $\hat{\mathcal{B}} \subseteq \mathcal{S}_i$ from a partitioning of $\mathcal{S}_i$ defined by $\hat{\mathcal{I}} = \mathcal{I}$ and $\hat{\mathcal{E}} = \mathcal{E}$ that satisfies $\hat{\mathcal{B}} = \mathcal{B} \cap \mathcal{S}_i$. Therefore, $\Lambda_{\hat{\mathcal{B}}} = \sum_{S \in \hat{\mathcal{B}}} \lambda(|S|) = \sum_{S \in \mathcal{B} \cap \mathcal{S}_i} \lambda(|S|)$. By Proposition 3.2, we have $\Lambda_{\hat{\mathcal{B}}} = \Lambda_{\mathcal{B}}$.

Overall, we obtain the bounds

$$\sum_{\mathcal{B} \in \mathsf{B}_{-i}^{(t)}} \Lambda_{\mathcal{B}}^+ \underline{v}_{\mathcal{B}} + \sum_{\mathcal{B} \in \mathsf{B}_{\pm i}^{(t)}} \Lambda_{\mathcal{B}} \underline{v}_{\mathcal{B}} \leq \sum_{S \in \mathcal{S}_i} \lambda(|S|)v(S) \leq \sum_{\mathcal{B} \in \mathsf{B}_{-i}^{(t)}} \Lambda_{\mathcal{B}}^+ \overline{v}_{\mathcal{B}} + \sum_{\mathcal{B} \in \mathsf{B}_{\pm i}^{(t)}} \Lambda_{\mathcal{B}} \overline{v}_{\mathcal{B}}.$$

Inserting these bounds into Equation (8) gives us Proposition 4.3, i.e.,

$$\varphi_i \geq \sum_{\mathcal{B} \in \mathsf{B}_{+i}^{(t)}} \Lambda_{\mathcal{B}}^- \underline{v}_{\mathcal{B}} + \sum_{\mathcal{B} \in \mathsf{B}_{\pm i}^{(t)}} \Lambda_{\mathcal{B}} \underline{v}_{\mathcal{B}} - \sum_{\mathcal{B} \in \mathsf{B}_{-i}^{(t)}} \Lambda_{\mathcal{B}}^+ \overline{v}_{\mathcal{B}} - \sum_{\mathcal{B} \in \mathsf{B}_{\pm i}^{(t)}} \Lambda_{\mathcal{B}} \overline{v}_{\mathcal{B}}$$

$$= \sum_{\mathcal{B} \in \mathsf{B}_{+i}^{(t)}} \Lambda_{\mathcal{B}}^- \underline{v}_{\mathcal{B}} + \sum_{\mathcal{B} \in \mathsf{B}_{\pm i}^{(t)}} \Lambda_{\mathcal{B}} (\underline{v}_{\mathcal{B}} - \overline{v}_{\mathcal{B}}) - \sum_{\mathcal{B} \in \mathsf{B}_{-i}^{(t)}} \Lambda_{\mathcal{B}}^+ \overline{v}_{\mathcal{B}} = \underline{\varphi}_i^{(t)}$$

$$\varphi_i \leq \sum_{\mathcal{B} \in \mathsf{B}_{+i}^{(t)}} \Lambda_{\mathcal{B}}^- \overline{v}_{\mathcal{B}} + \sum_{\mathcal{B} \in \mathsf{B}_{\pm i}^{(t)}} \Lambda_{\mathcal{B}} \overline{v}_{\mathcal{B}} - \sum_{\mathcal{B} \in \mathsf{B}_{-i}^{(t)}} \Lambda_{\mathcal{B}}^+ \underline{v}_{\mathcal{B}} - \sum_{\mathcal{B} \in \mathsf{B}_{\pm i}^{(t)}} \Lambda_{\mathcal{B}} \underline{v}_{\mathcal{B}}$$

$$= \sum_{\mathcal{B} \in \mathsf{B}_{+i}^{(t)}} \Lambda_{\mathcal{B}}^- \overline{v}_{\mathcal{B}} + \sum_{\mathcal{B} \in \mathsf{B}_{\pm i}^{(t)}} \Lambda_{\mathcal{B}} (\overline{v}_{\mathcal{B}} - \underline{v}_{\mathcal{B}}) - \sum_{\mathcal{B} \in \mathsf{B}_{-i}^{(t)}} \Lambda_{\mathcal{B}}^+ \underline{v}_{\mathcal{B}} = \overline{\varphi}_i^{(t)}.$$

$\square$

## A.3. Termination

**Theorem 3.3** (Restated). *There exists $t \in \mathbb{N}$, such that $\underline{\varphi_i}^{(t)} = \overline{\varphi_i}^{(t)} = \varphi_i$.*

*Proof.* To show that VERISHAP must eventually terminate, we note that in iteration $t = 2^{n-1}$, each branch contains only a single set of features $S \in \mathcal{S}_i$. Since we compute $[\underline{\Delta_{i\mathcal{B}}}, \overline{\Delta_{i\mathcal{B}}}]$ using bound propagation, by the assumptions we pose in Section 2.2 on bound propagation methods, for each branch $\mathcal{B} \in \mathrm{B}^{(t)}$ we have $\underline{\Delta_{i\mathcal{B}}} = \overline{\Delta_{i\mathcal{B}}} = \Delta_i(S)$ where $\mathcal{B} = \{S\}$. Now, as a direct result of Theorem 3.1, we have $\underline{\varphi_i}^{(t)} = \overline{\varphi_i}^{(t)} = \varphi_i$. Hence, VERISHAP terminates with $\underline{\varphi_i}^{(2^{n-1})} = \overline{\varphi_i}^{(2^{n-1})} = \varphi_i$.

$\square$

**Corollary 3.4** (Restated). *Given any precision level $\delta \in \mathbb{R}_{\geq 0}$, there exists $t \in \mathbb{N}$, such that $\overline{\varphi_i}^{(t)} - \underline{\varphi_i}^{(t)} \leq \delta$. This implies $|\hat{\varphi}_i - \varphi_i| \leq \delta$ for every $\hat{\varphi}_i \in [\underline{\varphi_i}^{(t)}, \overline{\varphi_i}^{(t)}]$.*

*Proof.* From Theorem 3.3 we know that the algorithm reaches the exact SHAP value at some iteration, hence it is also guaranteed that for any given precision tolerance $\delta \in \mathbb{R}_{\geq 0}$, there exists some iteration over which the computed bounds satisfy $\overline{\varphi_i}^{(t)} - \underline{\varphi_i}^{(t)} \leq \delta$. Hence, at this iteration, we can terminate the algorithm and return the computed bounds up to this precision.

$\square$

**Proposition 3.5** (Restated). *If $[\underline{\Delta}_i, \overline{\Delta}_i]$ are computed using an LBP method and $f(\mathbf{x}) = \mathbf{w}^\mathsf{T}\mathbf{x} + \mathbf{b}$, it holds that $\underline{\varphi_i}^{(1)} = \overline{\varphi_i}^{(1)} = \varphi_i$.*

*Proof.* Let $f(\mathbf{x}) = \mathbf{w}^\mathsf{T}\mathbf{x} + \mathbf{b}$. We follow the arguments provided by Lundberg & Lee (2017, Corollary 1) to show that the contribution $\Delta_i(S)$ is constant, or, in other words, the value of $\Delta_i(S)$ is independent of $S$. First, by the linearity of $f$, we obtain $v(S) = \mathbf{w}^\mathsf{T}\mathbb{E}_{\mathbf{z} \in \mathcal{D}}[\mathbf{x}_S; \mathbf{z}_{\bar{S}}] + \mathbf{b}$, where $[\mathbf{x}_S; \mathbf{z}_{\bar{S}}]$ denotes a vector where the features in $S$ are taken from the corresponding features in $\mathbf{x}$ and those in $\bar{S} = [n] \setminus S$ are taken from $\mathbf{z}$. In consequence,

$$
\begin{aligned}
\Delta_i(S) &= v(S \cup \{i\}) - v(S) \\
&= \mathbf{w}^\mathsf{T}\mathbb{E}_{\mathbf{z} \in \mathcal{D}}[\mathbf{x}_{S \cup \{i\}}; \mathbf{z}_{\overline{S \cup \{i\}}}] + \mathbf{b} - \mathbf{w}^\mathsf{T}\mathbb{E}_{\mathbf{z} \in \mathcal{D}}[\mathbf{x}_S; \mathbf{z}_{\bar{S}}] - \mathbf{b} \\
&= \mathbf{w}^\mathsf{T}(\mathbb{E}_{\mathbf{z} \in \mathcal{D}}[\mathbf{x}_{S \cup \{i\}}; \mathbf{z}_{\overline{S \cup \{i\}}}] - \mathbb{E}_{\mathbf{z} \in \mathcal{D}}[\mathbf{x}_S; \mathbf{z}_{\bar{S}}]) \\
&= \sum_{j=1}^{n} \mathbf{w}_j(\mathbb{E}_{\mathbf{z} \in \mathcal{D}}[\mathbf{x}_{S \cup \{i\}}; \mathbf{z}_{\overline{S \cup \{i\}}}]_j - \mathbb{E}_{\mathbf{z} \in \mathcal{D}}[\mathbf{x}_S; \mathbf{z}_{\bar{S}}]_j) \\
&= \mathbf{w}_i(\mathbf{x}_i - \mathbb{E}_{\mathbf{z} \in \mathcal{D}}\mathbf{z}_i).
\end{aligned}
$$

By propagating linear functions as bounds, LBP methods do not suffer from approximation error for compositions of linear functions. Since $\Delta_i$ is a composition of linear functions, LBP produces $\underline{\Delta_{i\mathcal{B}}} = \overline{\Delta_{i\mathcal{B}}} = \mathbf{w}_i(\mathbf{x}_i - \mathbb{E}_{\mathbf{z} \in \mathcal{D}}\mathbf{z}_i)$ for any branch $\mathcal{B}$, including the initial branch $\mathcal{B}_1^{(1)} = \mathcal{S}_i$. Therefore,

$$
\underline{\varphi_i}^{(1)} = \Lambda_{\mathcal{B}_1^{(1)}}\underline{\Delta_i}_{\mathcal{B}_1^{(1)}} = \mathbf{w}_i(\mathbf{x}_i - \mathbb{E}_{\mathbf{z} \in \mathcal{D}}\mathbf{z}_i) = \Lambda_{\mathcal{B}_1^{(1)}}\overline{\Delta_i}_{\mathcal{B}_1^{(1)}} = \overline{\varphi_i}^{(1)},
$$

and since $\varphi_i = \mathbf{w}_i(\mathbf{x}_i - \mathbb{E}_{\mathbf{z} \in \mathcal{D}}\mathbf{z}_i)$ (Lundberg & Lee, 2017, Corollary 1), this proves that $\underline{\varphi_i}^{(1)} = \overline{\varphi_i}^{(1)} = \varphi_i$.

$\square$

# B. Splitting Strategies

This section provides additional details on the splitting strategies introduced in Section 5. Appendix E.1 provides an experimental comparison of both the SELECT and SPLIT strategies presented below.

**Selecting Branches to Split (SELECT).** The partitioning strategy in Section 3 requires selecting a branch $\mathcal{B}_k$ to split. Since we process branches in batches, we instead select a batch $\mathcal{B}_{k_1}, \ldots, \mathcal{B}_{k_b}$ of branches to split simultaneously, where $b \in \mathbb{N}$

is the batch size. We present two strategies for selecting branches to split: MAXDIAM selects the $b$ branches $\mathcal{B}$ in $\mathsf{B}^{(t)}$, where $d_{\mathcal{B}} := \Lambda_{\mathcal{B}}(\overline{\Delta}_{i_{\mathcal{B}}} - \underline{\Delta}_{i_{\mathcal{B}}})$ is the largest, while MINDIAM selects the $b$ branches with the smallest $d_{\mathcal{B}}$. The motivation behind MAXDIAM is to greedily select those branches that contribute the most to the gap between $\underline{\varphi_i}$ and $\overline{\varphi_i}$. Conversely, MINDIAM seeks to select those branches first that are likely to be pruned soon, in order to reduce the memory demands of VERISHAP.

**Selecting Features to Split on (SPLIT).**  The branch-and-bound literature provides numerous strategies for selecting features to split, which can also be applied in VERISHAP. We adapt strong branching (Applegate et al., 1995), smart branching (Bunel et al., 2020; Boetius et al., 2025), and smears (Kearfott, 1996; Wang et al., 2018a) to our setting and compare the strategies experimentally in Appendix E.1. In the following, let $\mathcal{B} \in \mathsf{B}^{(t)}$ be defined by $\mathcal{I}, \mathcal{E} \subseteq [n] \setminus \{i\}$. We select a feature $j^*$ to split from the set of features $\mathcal{J} := [n] \setminus (\{i\} \cup \mathcal{I} \cup \mathcal{E})$ that are available for splitting.

- **INORDER.** A naïve approach for selecting features to split is selecting the features in a predefined order, e.g., $j^* := \min \mathcal{J}$. INORDER provides a baseline for evaluating other strategies.

- **STRONGBRANCHING.** This more powerful splitting strategy greedily selects those features that lead to the smallest diameter of the bounds on the contribution $\underline{\Delta}_{i_{\mathcal{B}}}, \overline{\Delta}_{i_{\mathcal{B}}}$ across the branches resulting from the split. Formally, if $[\underline{\Delta}_{i_j}', \overline{\Delta}_{i_j}']$ and $[\underline{\Delta}_{i_j}'', \overline{\Delta}_{i_j}'']$ are the contribution bounds of the two branches resulting from splitting feature $j$, STRONGBRANCHING selects $j^* := \arg \min_j \max(\overline{\Delta}_{i_j}' - \underline{\Delta}_{i_j}', \overline{\Delta}_{i_j}'' - \underline{\Delta}_{i_j}'')$ for splitting. STRONGBRANCHING is a comparatively expensive strategy, since it has to compute bounds on the contribution for each possible split.

- **IBPSTRONGBRANCHING.** This splitting strategy aims to approximate STRONGBRANCHING, while reducing its computational cost. It is an instance of a *smart branching* strategy. In particular, IBPSTRONGBRANCHING applies the smart branching strategy of Boetius et al. (2025) that evaluates all possible splits in the same manner as STRONGBRANCHING, but uses a cheaper bound propagation method (IBP) for bounding the contributions.

- **SMEARS.** The magnitude of the contribution gradient $\nabla \Delta_i(S) \in \mathbb{R}^{n-1}$ gives an indication along which features the contribution varies most. SMEARS leverages this information for determining which feature to split. For the branch $\mathcal{B}$, SMEARS selects $j^* := \arg \max_j \overline{|\nabla \Delta_i|}_j$, where $\overline{|\nabla \Delta_i|} \geq |\nabla \Delta_i(S)|$ is computed using IBP.

## C. Additional Details on VERISHAP

This section complements Sections 4 and 5 and Appendix B by providing further details on *(i)* relaxing the Boolean masks $\mathbf{m}$ to bounds over a continuous domain $[\underline{\mathbf{m}}, \overline{\mathbf{m}}]$, and *(ii)* performing bound propagation on the marginal value function. For ease of exposition, we describe these aspects for computing bounds on a single SHAP value $\varphi_i$, while our discussion extends naturally to simultaneously bounding all SHAP values $\varphi_1, \ldots, \varphi_n$, as described in Section 4.

**Relaxing Boolean Masks to Bounds.**  As described in Section 4, we represent sets of features, or coalitions, $S \in \mathcal{S}_i$ using Boolean masks $\mathbf{m} \in \{0, 1\}^n$ for computing bounds on the contribution $\Delta_i : \mathcal{S}_i \to \mathbb{R}$. We represent branches $\mathcal{B} \subseteq \mathcal{S}_i$ using bounds on the Boolean masks, i.e. $[\underline{\mathbf{m}}, \overline{\mathbf{m}}] \subseteq \mathbb{R}^n$, where $\underline{\mathbf{m}}, \overline{\mathbf{m}} \in \{0, 1\}^n$. Performing bound propagation on $\Delta_i$ given $[\underline{\mathbf{m}}, \overline{\mathbf{m}}]$ corresponds to relaxing the Boolean masks to the continuous domain $[0, 1]^n$. For this purpose, we define $\boldsymbol{\Delta}_i : [0, 1]^n \to \mathbb{R}$ so that $\boldsymbol{\Delta}_i(\mathbf{m}) = \Delta_i(S)$ if $\mathbf{m}$ corresponds to $S$.

Most operations performed by $\Delta_i$ can equally be applied by $\boldsymbol{\Delta}_i$. However, $\Delta_i$ also relies on computing $S \cup \{i\}$ in Equation (2) and the vector $(\mathbf{x}_S; \mathbf{z}_{\bar{S}})$ that is part of the marginal value function. We now define equivalent operations for Boolean masks that are used by $\boldsymbol{\Delta}_i$ in place of these operations. Let $\mathbf{m}$ be the mask vector corresponding to $S$. First, for $S \cup \{i\}$, we insert a 1 into $\mathbf{m}$ at index $i$ to obtain $\mathbf{m}^{+i}$, which corresponds to $S \cup \{i\}$. This is a linear operation that is similarly applicable to real-valued vectors. Second, we use that we already identified the Booleans with $\{0, 1\}$, so that $(\mathbf{x}_S; \mathbf{z}_{\bar{S}}) = \mathbf{m} * \mathbf{x} + (1 - \mathbf{m}) * \mathbf{z}$, where $*$ denotes element-wise multiplication. Overall, we have $\boldsymbol{\Delta}_i(\mathbf{m}) = \mathbf{v}(\mathbf{m}^{+i}) - \mathbf{v}(\mathbf{m})$, where $\mathbf{v}(\mathbf{m}) = \mathbb{E}_{\mathbf{z} \sim \mathcal{D}}[f(\mathbf{m} * \mathbf{x} + (1 - \mathbf{m}) * \mathbf{z})]$. Therefore, by relaxing the Boolean masks to $[0, 1]^{n-1}$, we also consider linear combinations of $\mathbf{x}$ and $\mathbf{z}$. However, since all operations besides $f$ are linear, this does not affect the tightness of linear bound propagation techniques as shown in Proposition 3.5.

**Propagating Bounds through the Marginal Value Function.** The marginal value function as defined in Section 2.1 is the expected value of the neural network output, typically over a dataset of background samples $\mathcal{D}$. We now discuss performing bound propagation on this expected value. Since, practically, $\mathcal{D}$ is a finite dataset, we have $v(S) = \mathbb{E}_{\mathbf{z}\sim\mathcal{D}}[f(\mathbf{x}_S; \mathbf{z}_{\bar{S}})] = 1/|\mathcal{D}| \sum_{\mathbf{z}\in\mathcal{D}} f(\mathbf{x}_S; \mathbf{z}_{\bar{S}})$, where $|\mathcal{D}|$ is the size of the background dataset. Since computing the average is a linear function of the network output, bound propagation proceeds as described in Section 2.2 for linear functions. Concretely, if $[\underline{f}_{\mathbf{z}}, \overline{f}_{\mathbf{z}}]$ are bounds on $f(\mathbf{x}_S; \mathbf{z}_{\bar{S}})$ for each $\mathbf{z}\in\mathcal{D}$, we have

$$\frac{1}{|\mathcal{D}|}\sum_{\mathbf{z}\in\mathcal{D}} \underline{f}_{\mathbf{z}} \le v(S) \le \frac{1}{|\mathcal{D}|}\sum_{\mathbf{z}\in\mathcal{D}} \overline{f}_{\mathbf{z}}.$$

## D. Datasets and Networks

Table 3 summarises the datasets and neural networks we use in our experiments.

**Neural Networks.** For all tabular datasets, we train fully connected neural networks, manually tuning the architecture to achieve maximal test set performance. On the `Mushroom` dataset, we train neural networks with different architectures to demonstrate the versatility of VERISHAP in Section 6.5. Additionally, we train neural networks of increasing size on `Mushroom` to test the scalability of VERISHAP with respect to the network size in Appendix E.3. The main `Mushroom` neural network used in our remaining experiments contains a single hidden layer of 8 ReLU neurons. We train convolutional neural networks on all vision datasets, using a standard convolutional architecture, and manually tune the network size to achieve good test set performance.

**Datasets.** All tabular datasets are taken from the UCI ML Repository (Kelly et al.), which provides concise descriptions of each dataset. `MNIST` is the standard, widely known hand-written digit recognition dataset introduced by LeCun et al. (1998). `FashionMNIST` (Xiao et al., 2017) is a more challenging dataset in the same format as `MNIST`, concerned with classifying images of fashion items. `CIFAR10` (Krizhevsky, 2009) is concerned with recognising vehicles and animals in $32 \times 32$ pixel colour images. `GTSRB` (Stallkamp et al., 2011) contains colour images of German traffic signs scaled to the same size as `CIFAR10`.

## E. Additional Experiments

This appendix contains additional experiments on different bound propagation algorithms, branch selection strategies, and split selection strategies for VERISHAP in Appendix E.1, quantifying the effect of different sources of bound looseness in VERISHAP in Appendix E.2, analysing the scalability of VERISHAP in the network size in Appendix E.3, further results on image datasets in Appendices E.4 and E.5, extended statistics for our comparison to EXACTSHAP in Appendix E.6, results for additional statistical SHAP estimators in Appendices E.7 and E.8, and an experiment on predicting the runtime of VERISHAP in Appendix E.9. Below, we first provide additional details on the value functions, summarise the metrics used in our experiments, and provide additional details on our hardware setup and our hyper-parameter values for VERISHAP.

**Value Functions.** Our experiments leverage three variants of the marginal value function introduced in Section 2.1. In Sections 6.2 and 6.3 and Appendices E.1 and E.8, we study the marginal value function $v_{\text{marginal}}(S) := \mathbb{E}_{\mathbf{z}\sim\mathcal{D}}[f(\mathbf{x}_S; \mathbf{z}_{\bar{S}})]$, where $\mathcal{D}$ is a background dataset of 100 samples. We additionally use the *baseline* value function $v_{\text{baseline}}(S) := f(\mathbf{x}_S; \mathbf{z}_{\bar{S}})$ for the *baseline input* $\mathbf{z} \in \mathbb{R}^n$ that corresponds to the marginal value function under a singleton background dataset. Concretely, we use the *zero-baseline* value function, where $\mathbf{z} = \mathbf{0}_n$, in Section 6.1 and Appendix E.5 and the *mean-baseline* value function, where $\mathbf{z}$ is the input mean over the training dataset, in Section 6.4 and Appendices E.5 and E.8.

**Evaluation Metrics.** We frequently compare SHAP bounds to the attributed neural network output to assess the tightness of the bounds. Given a multi-class classifier or multi-regressor $f : \mathbb{R}^n \to \mathbb{R}^m$, one output $k \in [m]$ is selected to be attributed. Given SHAP bounds $[\underline{\varphi_i}, \overline{\varphi_i}]$, we evaluate, for example, whether the bounds half-range $(\overline{\varphi_i} - \underline{\varphi_i})/2$ is at most $p\%$ of the network output to attribute $f(\mathbf{x})_k$ ('$p\%$ HR'). Furthermore, we rely on the maximum ($\ell_\infty$) error between estimated SHAP values $\hat{\varphi}$ and the exact SHAP values $\varphi$, which is $\max_i |\hat{\varphi}_i - \varphi_i|$. For better comparability with Witter et al. (2025), we also report the mean squared error $\frac{1}{n}\|\hat{\varphi} - \varphi\|_2^2$.

*Table 3.* **Datasets and Neural Networks used in our Experiments.** 'Architecture' lists the neural network used on each dataset, specified as the sequence of layers. We use $FC_k^\sigma$ to denote a $\sigma$-activated fully-connected layer of $k$ neurons. If $\sigma$ is omitted, the layer is ReLU-activated. Similarly, $Out_m$ denotes the linear output layer of dimension $m$, $Conv_c(k_1 \times k_2)$ denotes a convolution layer with $c$ channels, a window size of $k_1 \times k_2$, a stride of one, and 'same' padding, $AvgPool(k_1 \times k_2)$ denotes an average pooling layer with a window size of $k_1 \times k_2$, strides of $k_1$ and $k_2$, and 'valid' padding, SkipConnection denotes a residual connection adding the input of the previous layer to the output of the layer, BN denotes batch normalisation, and ReLU denotes a ReLU activation layer for convolutional neural networks. For classification tasks, 'Performance' specifies accuracy. For regression tasks, 'Performance' provides the root mean square error.

| $n$ | Dataset | Architecture | Task | Performance Train | Test |
|---|---|---|---|---|---|
| 12 | Adult (Becker & Kohavi, 1996) | $FC_{32}, FC_{32}, Out_2$ | Classification | 85% | 85% |
| 16 | Obesity (Palechor & de la Hoz Manotas, 2019) | $FC_{32}, FC_{32}, Out_7$ | Classification | 100% | 95% |
| 20 | German (Hofmann, 1994) | $FC_8, Out_1$ | Classification | 81% | 77% |
| 23 | Default (Yeh & Lien, 2009) | $FC_{64}, FC_{64}, FC_{64}, Out_1$ | Classification | 90% | 71% |
| 25 | Auto (Schlimmer, 1985) | $FC_{32}, FC_{32}, Out_1$ | Regression | 0.14 | 0.59 |
| 27 | Steel (Buscema et al., 2010) | $FC_8, FC_8, Out_7$ | Classification | 78% | 72% |
| 28 | HepatitisC (Nasr et al., 2017) | $FC_8, FC_8, Out_4$ | Classification | 53% | 25% |
| 30 | BreastCancer (Street et al., 1993) | $FC_{32}, FC_{32}, Out_1$ | Classification | 100% | 97% |
| 38 | Annealing (Annealing Dataset) | $FC_{32}, FC_{32}, Out_1$ | Classification | 100% | 99% |
| 56 | LungCancer (Hong & Yang, 1991) | $FC_4, FC_4, Out_3$ | Classification | 84% | 57% |
| 58 | Online News (Fernandes et al., 2015) | $FC_{64}, FC_{64}, Out_1$ | Classification | 67% | 66% |
| 60 | Sonar (Sejnowski & Gorman, 1988) | $FC_{32}, FC_{32}, Out_1$ | Classification | 100% | 100% |
| 22 | Mushroom (Mushroom) | $FC_1, Out_1$ | Classification | 96% | 95% |
| 22 | Mushroom (Mushroom) | $FC_2, Out_1$ | Classification | 96% | 96% |
| 22 | Mushroom (Mushroom) | $FC_4, Out_1$ | Classification | 99% | 99% |
| 22 | Mushroom (Mushroom) | $FC_8, Out_1$ | Classification | 100% | 100% |
| 22 | Mushroom (Mushroom) | $FC_{16}, Out_1$ | Classification | 100% | 100% |
| 22 | Mushroom (Mushroom) | $FC_{32}, Out_1$ | Classification | 100% | 100% |
| 22 | Mushroom (Mushroom) | $FC_{64}, Out_1$ | Classification | 100% | 100% |
| 22 | Mushroom (Mushroom) | $FC_{128}, Out_1$ | Classification | 100% | 100% |
| 22 | Mushroom (Mushroom) | $FC_{256}, Out_1$ | Classification | 100% | 100% |
| 22 | Mushroom (Mushroom) | $FC_{512}, Out_1$ | Classification | 100% | 100% |
| 22 | Mushroom (Mushroom) | $FC_{1024}, Out_1$ | Classification | 100% | 100% |
| 22 | Mushroom (Mushroom) | $FC_{2048}, Out_1$ | Classification | 100% | 100% |
| 22 | Mushroom (Mushroom) | $FC_{4096}, Out_1$ | Classification | 100% | 100% |
| 22 | Mushroom (Mushroom) | $FC_{8192}, Out_1$ | Classification | 100% | 100% |
| 22 | Mushroom (Mushroom) | $FC_{16384}, Out_1$ | Classification | 100% | 100% |
| 22 | Mushroom (Mushroom) | $FC_{32768}, Out_1$ | Classification | 100% | 100% |
| 22 | Mushroom (Mushroom) | $FC_8^{\tanh}, Out_1$ | Classification | 98% | 98% |
| 22 | Mushroom (Mushroom) | $FC_8^{\text{swish}}, Out_1$ | Classification | 100% | 100% |
| 22 | Mushroom (Mushroom) | ResNet* | Classification | 100% | 100% |
| 784 | MNIST (LeCun et al., 1998) | ConvNet† | Classification | 98.9% | 98.6% |
| 784 | FashionMNIST (Xiao et al., 2017) | ConvNet† | Classification | 97.0% | 89.0% |
| 3072 | CIFAR10 (Krizhevsky, 2009) | ConvNet‡ | Classification | 87.3% | 68.7% |
| 3072 | GTSRB (Stallkamp et al., 2011) | ConvNet⋆ | Classification | 100% | 90.5% |

*$FC_{22}$, SkipConnection, $FC_{22}$, SkipConnection, $FC_{22}$, SkipConnection, $Out_1$

†$Conv_4(5 \times 5)$, $AvgPool(2 \times 2)$, BN, ReLU, $Conv_8(5 \times 5)$, $AvgPool(2 \times 2)$, BN, ReLU, $FC_{392}$, $FC_{64}$, $Out_{10}$

‡$Conv_{16}(5 \times 5)$, $AvgPool(2 \times 2)$, BN, ReLU, $Conv_{32}(5 \times 5)$, $AvgPool(2 \times 2)$, BN, ReLU, $FC_{2048}$, $FC_{256}$, $Out_{10}$

⋆$Conv_{32}(5 \times 5)$, $AvgPool(2 \times 2)$, BN, ReLU, $Conv_{64}(5 \times 5)$, $AvgPool(2 \times 2)$, BN, ReLU, $FC_{4096}$, $FC_{256}$, $Out_{10}$

**Hardware & Hyperparameters.** Our main hardware is an L40S NVIDIA GPU with 48GB of GPU memory, deployed on an Ubuntu 24.04 machine with 188GB of RAM and an AMD EPYC 9375F 32-Core processor. Unless otherwise noted, we run VERISHAP with a batch size of 4096.

### E.1. Ablation Studies

We compare different bound propagation algorithms, branch selection strategies, and split selection strategies. For this comparison, we apply VERISHAP to marginal SHAP with 100 background data points on a selection of the datasets from Section 6.2. Concretely, we use the German, Mushroom, Default, Auto, Steel, and Sonar datasets. In this experiment, we compute SHAP values only for the first test sample and set a timeout of 400s.

**Bound Propagation Algorithms.** We compare the IBP (Moore et al., 2009), CROWN (Zhang et al., 2018), CROWN-IBP (Zhang et al., 2020), and $\alpha$-CROWN (Xu et al., 2021) bound propagation algorithms for computing bounds on the value function in VERISHAP. In this comparison, we use the MAXDIAM branch selection and SMEARS split selection strategies. Due to the increased memory requirements of CROWN for recursively propagating bounds, CROWN exceeds the memory of our GPU for all networks we use in this comparison. For this reason, CROWN is excluded from the remainder of this comparison.

Table 4 contains the results of this comparison. IBP is fastest for computing exact SHAP values, but reaches the timeout or requires more runtime to compute tight bounds on the higher-dimensional datasets. While $\alpha$-CROWN computes the tightest contribution bounds, it is more runtime-intensive than CROWN-IBP and IBP, causing it to require more runtime both for computing exact SHAP values and tight bounds. Furthermore, the additional runtime cost of $\alpha$-CROWN means that it cannot perform as many iterations as CROWN-IBP and IBP within the timeout. For this reason, the final bounds that VERISHAP computes on the Steel and Sonar datasets are less tight when using $\alpha$-CROWN than for CROWN-IBP, while still significantly tighter than for IBP. Concretely, on Steel, the half-range of the SHAP value bounds is $1.52\%$ of the network output to attribute when using CROWN-IBP, $1.95\%$ for $\alpha$-CROWN, and $17.70\%$ for IBP. On Sonar, the half-range is $1.13\%$ for CROWN-IBP, $1.37\%$ for $\alpha$-CROWN, and $28.77\%$ for IBP. Overall, the bound propagation algorithm CROWN-IBP that we apply in our experiments performs best for computing tight bounds for higher-dimensional datasets.

**Branch Selection.** Another central component of VERISHAP is the strategy used to select the branches to split. We compare the MAXDIAM and MINDIAM strategies introduced in Section 5, while using CROWN-IBP as the bound propagation algorithm and SMEARS for selecting splits.

The results of our comparison are summarised in Table 5. While MINDIAM reduces the runtime for computing exact SHAP values, it increases the runtime required to obtain tight bounds compared to MAXDIAM. Since the inherent computational cost of MINDIAM and MAXDIAM is the same, the runtime differences we observe for computing exact SHAP values require further investigation, which we leave for future work. In our experiments, we rely on MAXDIAM, as it enables VERISHAP to compute tight bounds faster on most datasets.

**Split Selection.** Finally, we compare the INORDER, SMEARS, IBPSTRONGBRANCHING, and STRONGBRANCHING split selection strategies introduced in Appendix B. For this comparison, we use CROWN-IBP as the bound propagation algorithm and MAXDIAM for selecting branches.

Table 6 summarises the results of this comparison. While INORDER allows VERISHAP to compute exact SHAP values the fastest in most cases, due to its minimal runtime requirements, INORDER does not allow for computing tight bounds on the Steel and Sonar datasets. Both IBPSTRONGBRANCHING and STRONGBRANCHING only produce results on the lower-dimensional German and Mushroom datasets, due to their high runtime requirements. SMEARS is the only strategy that allows for computing tight bounds on the Steel and Sonar datasets. Furthermore, it offers a comparable runtime to INORDER for computing exact SHAP values. For this reason, we apply SMEARS in our experiments.

### E.2. Impact of Continuous Relaxation

As described in Section 4, VERISHAP relaxes discrete masks to a continuous space in order to apply bound propagation. Therefore, the bounds computed by VERISHAP can be loose for two reasons: the bound propagation and the continuous relaxation. In this section, we tell the two sources of looseness apart by computing the true range of the baseline value

*Table 4.* **Bound Propagation Algorithms.** This table contains the runtime required by VERISHAP for computing SHAP values ('Exact') or the bounds half-range reaching 10% of the network output to attribute ('10% HR'). A '–' denotes VERISHAP reaching the timeout, which is 400s for this experiment.

| Dataset | Exact Runtime | | | 10% HR Runtime | | |
|---|---|---|---|---|---|---|
| | $\alpha$-**CROWN** | **CROWN-IBP** | **IBP** | $\alpha$-**CROWN** | **CROWN-IBP** | **IBP** |
| German | 22s | 20s | **14s** | 18s | 16s | **12s** |
| Mushroom | 28s | 25s | **19s** | 19s | 16s | **15s** |
| Default | 138s | 132s | **92s** | 138s | 132s | **92s** |
| Auto | 317s | 315s | **257s** | 68s | **52s** | 74s |
| Steel | – | – | – | 114s | **86s** | – |
| Sonar | – | – | – | 16s | **13s** | – |

*Table 5.* **Branch Selection Strategies.** This table contains the runtime required by VERISHAP for computing SHAP values ('Exact') or the bounds half-range reaching 10% of the network output to attribute ('10% HR'). A '–' denotes VERISHAP reaching the timeout, which is 400s for this experiment.

| Dataset | Exact Runtime (s) | | 10% HR Runtime (s) | |
|---|---|---|---|---|
| | **MAXDIAM** | **MINDIAM** | **MAXDIAM** | **MINDIAM** |
| German | 20s | **18s** | **16s** | 18s |
| Mushroom | 25s | **21s** | **16s** | 21s |
| Default | 133s | **110s** | 133s | **110s** |
| Auto | 315s | **190s** | **52s** | 190s |
| Steel | – | – | **86s** | – |
| Sonar | – | – | 13s | 13s |

*Table 6.* **Split Selection Strategies.** This table contains the runtime required by VERISHAP for computing SHAP values ('Exact') or the bounds half-range reaching 10% of the network output to attribute ('10% HR'). A '–' denotes VERISHAP reaching the timeout, which is 400s for this experiment. In this table, 'ISTB' and 'STB' abbreviate IBPSTRONGBRANCHING and STRONGBRANCHING, respectively.

| Dataset | Exact Runtime (s) | | | | 10% HR Runtime (s) | | | |
|---|---|---|---|---|---|---|---|---|
| | **INORDER** | **SMEARS** | **ISTB** | **STB** | **INORDER** | **SMEARS** | **ISTB** | **STB** |
| German | **18s** | 20s | 21s | 28s | **15s** | 16s | 16s | 19s |
| Mushroom | 34s | **26s** | 48s | 77s | 25s | **16s** | 34s | 50s |
| Default | **111s** | 133s | – | – | **111s** | 133s | – | – |
| Auto | **289s** | 314s | – | – | 132s | **52s** | – | – |
| Steel | – | – | – | – | – | **87s** | – | – |
| Sonar | – | – | – | – | – | **13s** | – | – |

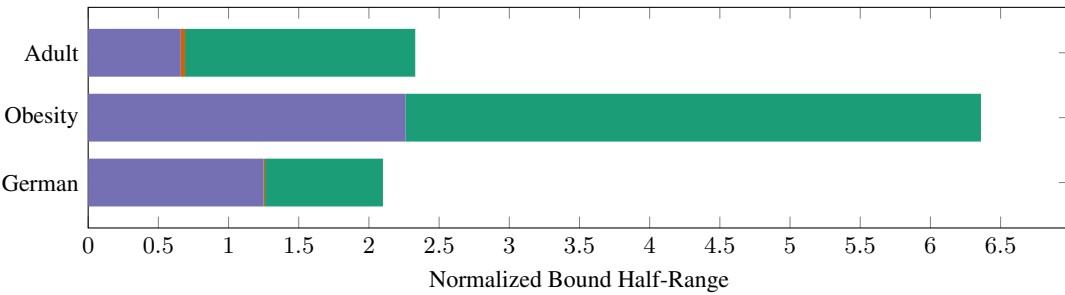

*Figure 6.* **Sources of Looseness in the VERISHAP Value Bounds.** This plot decomposes the initial value function $v$ bounds computed by VERISHAP into the true range of $v$ over the discrete coalition space ■, the looseness introduced by the continuous relaxation ■, and the looseness introduced by CROWN-IBP ■. For each dataset, we normalise the bounds by the network output to attribute. Since the looseness introduced by the continuous relaxation is marginal compared to the CROWN-IBP bounds for all three datasets, the ■ bar is barely visible in this plot.

function $v$ over the discrete coalition space and the relaxed continuous space on low-dimensional datasets. Comparing these ranges to the bounds computed by bound propagation reveals which technique is the larger source of looseness. We apply neural network verification based on mixed integer linear programming (MILP) to compute the true range over the relaxed continuous space. For the discrete space, we enumerate all coalitions to compute the true range. Figure 6 reveals that the contribution of the continuous relaxation is marginal across the `Adult`, `Obesity`, and `German` datasets.

### E.3. Scalability in Network Size

In this section, we investigate the scalability of VERISHAP with respect to the size of the neural network for which we compute SHAP values. For this experiment, we train neural networks of increasing size on the `Mushroom`. The networks are described in Appendix D. To reduce computational demand, we use the mean-baseline value function and compute SHAP values only for the first test sample in this experiment. To still obtain reliable runtime measurements, we repeat each run five times.

Figure 7 summarises the results of our comparison. As apparent from this figure, the runtime of VERISHAP scales *linearly* with the network size, measured by the number of network parameters, on the `Mushroom`. This perhaps surprising result is consistent with earlier observations in the related field of probabilistic verification of neural networks (Boetius et al., 2025). A key observation is that the network complexity determines the runtime of VERISHAP only indirectly, via the complexity of the decision boundary. For example, as Proposition 3.5 proves, VERISHAP terminates immediately also for high-dimensional linear functions with many parameters, since no splitting is required for computing tight bounds on the contribution function. As most networks used here achieve identical performance on `Mushroom`, they likely share similar decision boundaries. Another indicator of the decision boundary's complexity is the number of iterations, i.e., splits, that VERISHAP performs to compute the exact SHAP values. As Figures 7(c) and 7(d) show, the only substantial jump in runtime from the smallest network containing only a single neuron to the next larger network containing two neurons is connected to a jump in the number of iterations of VERISHAP, which indicates changes in the complexity of the decision boundary.

### E.4. Additional Statistics for Section 6.1

We provide additional diagnostic statistics for our MNIST experiment presented in Section 6.1 and Figure 2. Figure 8 complements Figure 3 by plotting the total number of branches, not currently pruned (active) branches, and pruned branches, as well as the reasons for pruning branches and the margin of the computed SHAP bounds. When inspected together with Figure 2, Figure 8 reveals that the SHAP bounds computed by VERISHAP become tight before any branches are pruned. Furthermore, all branches are pruned because they have tight value function bounds, rather than because they are fully split. The minimum and average bounds margin differ only minimally from the maximal bounds margin presented in Figure 8(c).

### E.5. Experiments on Additional Image Datasets

Complementing the results presented in Section 6.1, we apply VERISHAP to compute SHAP values on additional image datasets using both the zero-baseline and the mean-baseline value functions. Concretely, we compute bounds on `MNIST`,

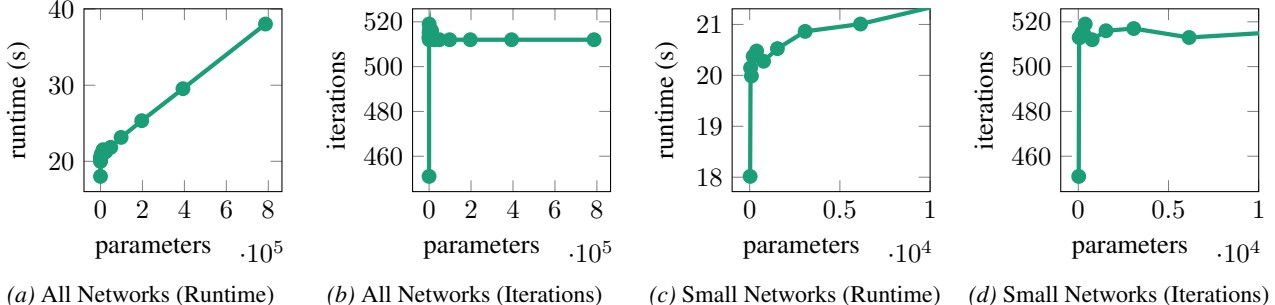

*(a)* All Networks (Runtime)  *(b)* All Networks (Iterations)  *(c)* Small Networks (Runtime)  *(d)* Small Networks (Iterations)

*Figure 7.* **Scalability in Network Size.** We plot the runtime and the number of iterations VERISHAP requires to compute the exact SHAP values for neural networks trained on the Mushroom dataset of increasing size. We measure network size by the number of parameters a network possesses. The right plots (c) and (d) show cut-outs of the left plots (a) and (b) focussing on the results for small networks.

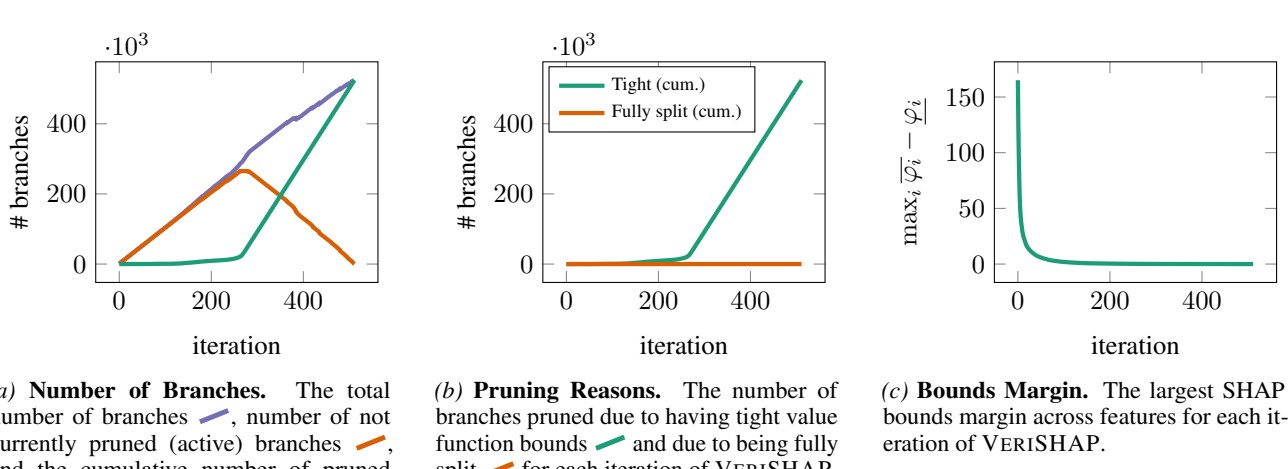

*(a)* **Number of Branches.** The total number of branches ⟋, number of not currently pruned (active) branches ⟋, and the cumulative number of pruned branches ⟋ for each iteration of VER-ISHAP.

*(b)* **Pruning Reasons.** The number of branches pruned due to having tight value function bounds ⟋ and due to being fully split ⟋ for each iteration of VERISHAP.

*(c)* **Bounds Margin.** The largest SHAP bounds margin across features for each iteration of VERISHAP.

*Figure 8.* **Extended MNIST Experiment Analysis.** This figure provides additional statistics for our MNIST experiment described in Section 6.1 and visualised in Figure 2

*Table 7.* **Extended Version of Table 1: VERISHAP vs. EXACTSHAP.** In addition to the median runtimes provided in Table 1, this table also provides the interquartile ranges ('IQR') of the runtimes.

| | | | VERISHAP (Ours) | | |
|---|---|---|---|---|---|
| | | **EXACT SHAP** | 10% HR | 1% HR | Exact |
| $\|\mathcal{P}([n])\|$ | **Dataset** | median (IQR) | median (IQR) | median (IQR) | median (IQR) |
| $2^{16}{}_{>10^4}$ | Obesity | 4s (4s–4s) | 18s (18s–18s) | 18s (18s–19s) | 18s (18s–19s) |
| $2^{20}{}_{>10^6}$ | German | 9s (9s–9s) | 16s (16s–17s) | 19s (17s–19s) | 20s (19s–20s) |
| $2^{22}{}_{>10^6}$ | Mushroom | – | 17s (16s–17s) | 20s (20s–32s) | 25s (25s–25s) |
| $2^{23}{}_{>10^6}$ | Default | – | 127s (123s–130s) | 132s (131s–133s) | 133s (132s–133s) |
| $2^{25}{}_{>10^7}$ | Auto | – | 81s (73s–85s) | 213s (187s–233s) | 316s (314s–317s) |
| $2^{27}{}_{>10^8}$ | Steel | – | 37s (30s–44s) | 322s (271s–374s) | – |
| $2^{30}{}_{>10^9}$ | BreastCancer | – | 49s (35s–67s) | 322s (271s–564s) | – |
| $2^{38}{}_{>10^{11}}$ | Annealing | – | 269s (124s–315s) | – | – |
| $2^{60}{}_{>10^{18}}$ | Sonar | – | 13s (13s–13s) | – | – |

*Table 8.* **SHAP Estimator Runtimes.** This table provides the runtime of 100 repetitions of different statistical estimators for the largest number of samples that does not exceed the GPU memory of our hardware, as plotted in Figure 11.

| | Runtime | | |
|---|---|---|---|
| **Estimator** | Mushroom | Default | Auto |
| KERNELSHAP | 346s | 123s | 351s |
| LEVERAGESHAP | 291s | 324s | 314s |
| LINEARMSR | 267s | 270s | 274s |
| TREEMSR | 895s | 132s | 341s |

FashionMNIST, CIFAR10, and GTSRB convolutional neural networks. Table 3 provides the architectures of these networks. For MNIST and FashionMNIST, we group the input images into evenly-spaced grids of superpixels, for which we compute SHAP values. On CIFAR10 and GTSRB, we compute superpixels using the SLIC (Achanta et al., 2012) and watershed (Neubert & Protzel, 2014) image segmentation algorithms. While we run VERISHAP with a batch size of 4096 until it reaches a timeout of 900s on MNIST and FashionMNIST, due to the size of the neural networks for CIFAR10 and GTSRB, we reduce the batch size to 1024 and increase the timeout to 2 hours. Figure 9 contains the results of VERISHAP using the mean-baseline value function. Figure 10 complements Figure 2 by presenting additional VERISHAP results for the zero-baseline value function.

### E.6. Additional Details on the Comparison to EXACTSHAP

Table 7 complements Table 1 by providing the interquartile ranges (IQRs) of the runtimes of VERISHAP and EXACTSHAP across the 10 test set samples we evaluate the algorithms on in Section 6.2. For up to 23 input dimensions, the IQRs are within 2s of the median runtime, with the exception of the Default dataset, where the IQR for VERISHAP computing '10% HR' bounds is 8s. For larger input spaces, the IQRs reach up to 191s for the Annealing dataset.

### E.7. Comparison to Additional Statistical Estimators

Section 6.3 compares VERISHAP to KERNELSHAP and TREEMSR. This section provides a comparison to the additional statistical estimators LEVERAGESHAP (Musco & Witter, 2025) and LINEARMSR (Witter et al., 2025). Figure 11 presents the results of our comparison of VERISHAP to these statistical estimators. Table 8 provides the runtime required for performing 100 repetitions of the different statistical estimators for the largest feasible sample size on our hardware. The additional results follow the same trend discussed for KERNELSHAP and TREEMSR in Section 6.3.

### E.8. Evaluating SHAP Estimators using VERISHAP

This section presents additional results on evaluating SHAP estimators using VERISHAP, as discussed in Section 6.4. Figure 12 extends Figure 5 by including additional datasets. In this section, we also include Steel in the comparison for which VERISHAP is unable to compute the exact SHAP values. Therefore, we report the *optimistic* mean squared

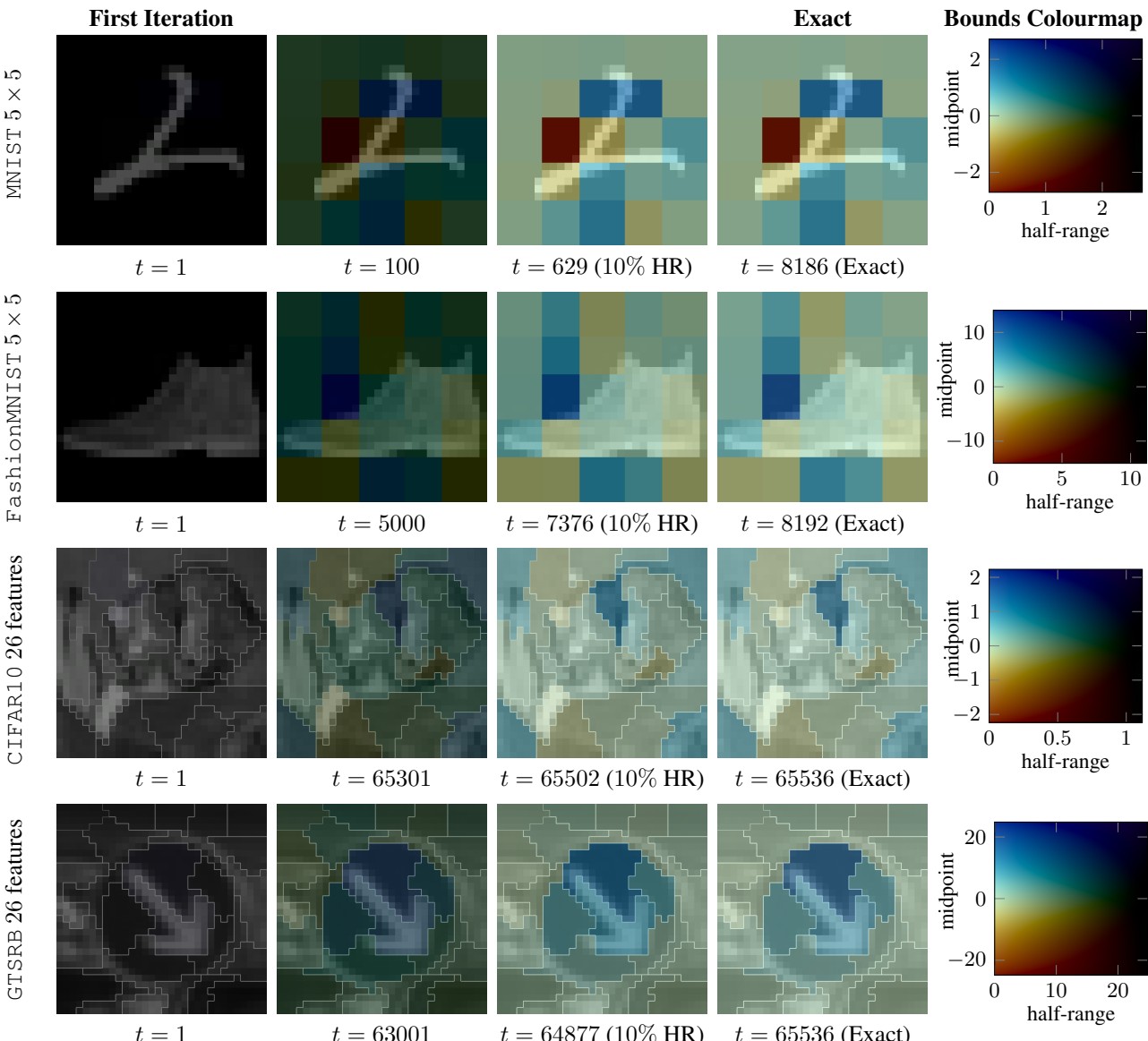

*Figure 9.* **VERISHAP Results for Mean-Baseline SHAP on MNIST, FashionMNIST, CIFAR10, & GTSRB.** For MNIST, we compute SHAP value bounds for the class '4' score on a test set image of a '2'. For FashionMNIST, we attribute the 'Ankle Boot' score on a test set image of an 'Ankle Boot'. For CIFAR10, we attribute the 'Cat' score on a test set image of a 'Cat'. For GTSRB, we attribute the 'Pass on the Right' score on a test set image of a 'Pass on the Right' sign. The SHAP bounds $\underline{\varphi_i}, \overline{\varphi_i}$ are visualised by their midpoints $(\overline{\varphi_i} + \underline{\varphi_i})/2$ and half-ranges (HR) $(\overline{\varphi_i} - \underline{\varphi_i})/2$ determining colour hue and lightness, respectively. **Lighter colour means tighter bounds in this figure.** We write '$p\%$ HR' to denote the iteration in which the largest half-range is less than $p\%$ of the network output for the output to attribute.

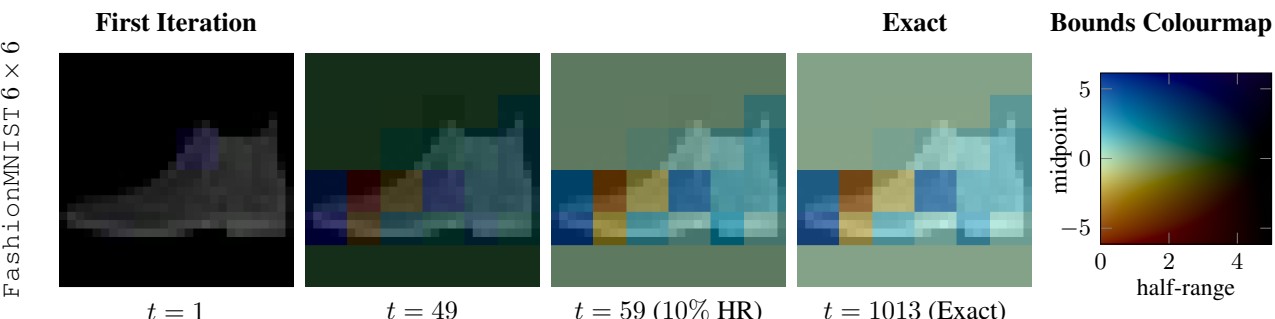

*Figure 10.* **VERISHAP Results for Zero-Baseline SHAP on FashionMNIST.** We attribute the 'Ankle Boot' score on a test set image of an 'Ankle Boot'. The SHAP bounds $\underline{\varphi_i}, \overline{\varphi_i}$ are visualised by their midpoints $(\overline{\varphi_i} + \underline{\varphi_i})/2$ and half-ranges (HR) $(\overline{\varphi_i} - \underline{\varphi_i})/2$ determining colour hue and lightness, respectively. **Lighter colour means tighter bounds in this figure.** We write '$p\%$ HR' to denote the iteration in which the largest half-range is less than $p\%$ of the network output for the output to attribute.

error $\frac{1}{n}\|\min(|\hat{\boldsymbol{\varphi}} - \underline{\boldsymbol{\varphi}}|, |\hat{\boldsymbol{\varphi}} - \overline{\boldsymbol{\varphi}}|)\|_2^2$ on Steel, where $\hat{\boldsymbol{\varphi}}$ are the estimated SHAP values and $[\underline{\boldsymbol{\varphi}}, \overline{\boldsymbol{\varphi}}]$ are the SHAP value bounds computed by VERISHAP.

The datasets in Figure 5 reflect the different relationships between TREEMSR, KERNELSHAP, and LEVERAGESHAP that we observe in Figure 12. In particular, the results on Obesity and Auto follow the same trend as on Mushroom, and the results on Steel follow the same trend as on German. Figure 13 compares LEVERAGESHAP to LINEARMSR (Witter et al., 2025), confirming the result of Witter et al. (2025) that LINEARMSR closely matches LEVERAGESHAP.

### E.9. Predicting VERISHAP Runtime

This section is concerned with predicting the runtime of VERISHAP before VERISHAP is run. The goal is to judge whether VERISHAP will successfully compute tight bounds for a neural network. Concretely, we study the number of branches required by VERISHAP until the half-width of the SHAP bounds is less than 10% of the network output to attribute (10% HR). The number of branches directly determines the runtime of VERISHAP while discounting for different network architectures, enabling better comparability across datasets. As Figure 14 shows, there is a strong correlation between the half-range of the initial SHAP bounds and the number of branches required for reaching 10% HR tight bounds. Therefore, the half-range of the initial bounds can be used to predict the time required until VERISHAP computes tight bounds.

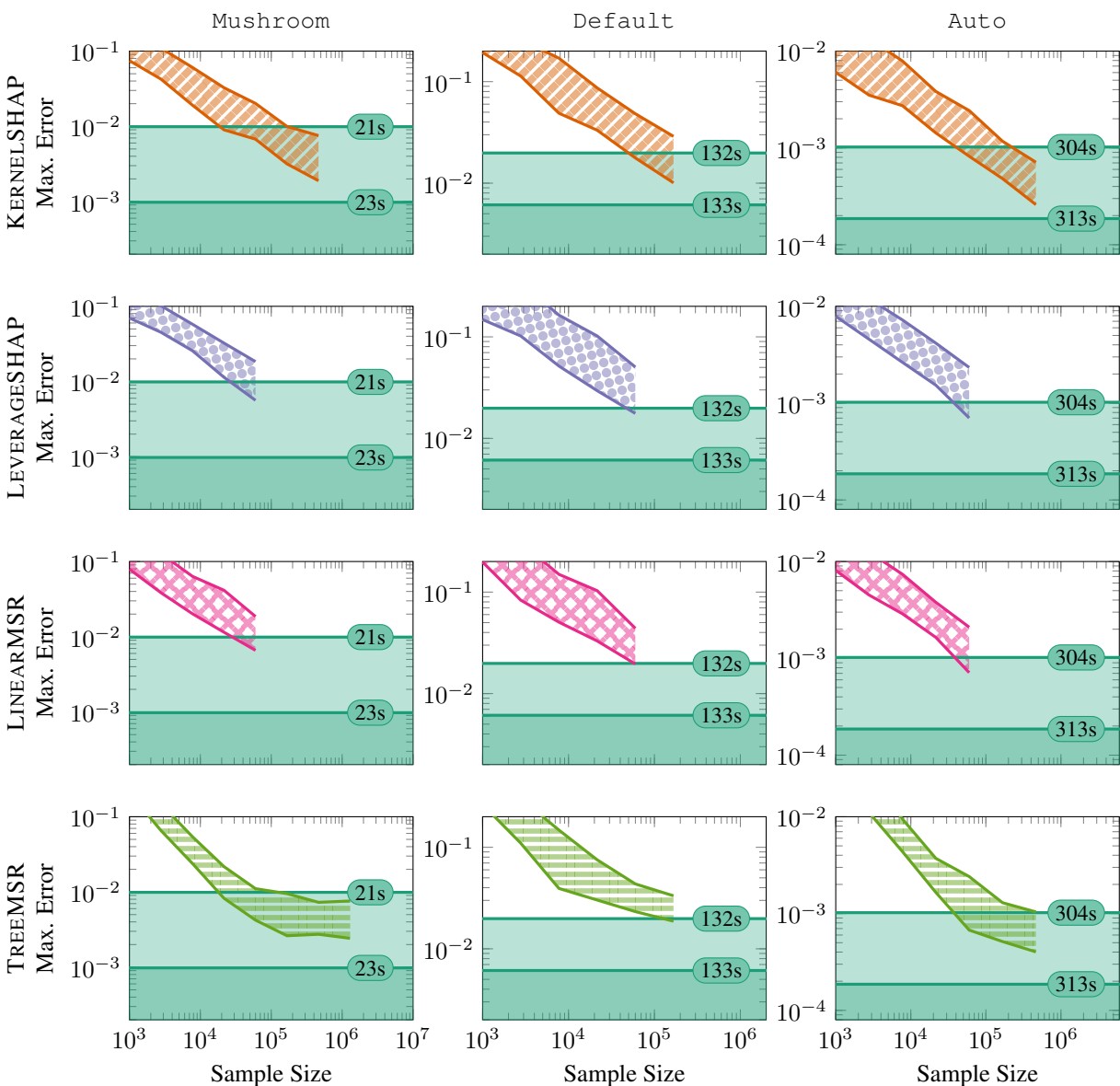

*Figure 11.* **VERISHAP** ■ **vs. KERNELSHAP** ▨, **LEVERAGESHAP** ▨, **LINEARMSR** ▨ **& TREEMSR** ▨. We run KERNELSHAP, LEVERAGESHAP, LINEARMSR, and TREEMSR with increasing sample sizes until each exhausts the available GPU memory, repeating each run 100 times. For each sample size, we plot the range across repetitions of the largest error between the estimated SHAP value and the true SHAP value of a feature. For VERISHAP, we mark the half-ranges of the SHAP bounds at different (runtimes).

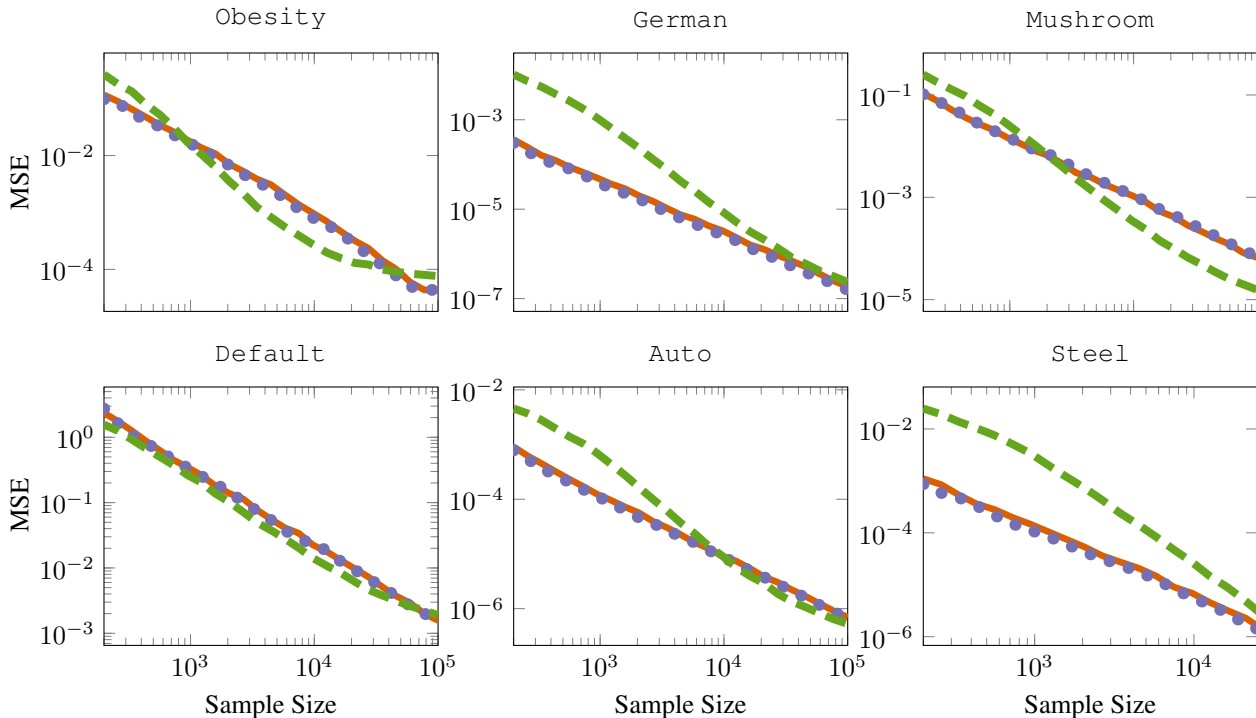

*Figure 12.* **KERNELSHAP** ━━ **vs. LEVERAGESHAP** •••• **vs. TREEMSR** ╌ ╌ **(Extended)**. We report the mean squared error (MSE) of statistical SHAP estimators across 100 runs as in Figure 5. Since the exact SHAP values are unknown for the `Steel`, we compute an optimistic MSE as $\frac{1}{n}\|\min(|\hat{\varphi} - \underline{\varphi}|, |\hat{\varphi} - \overline{\varphi}|)\|_2^2$, where $\hat{\varphi}$ are the estimated SHAP values and $[\underline{\varphi}, \overline{\varphi}]$ are the bounds on the exact SHAP values computed by VERISHAP.

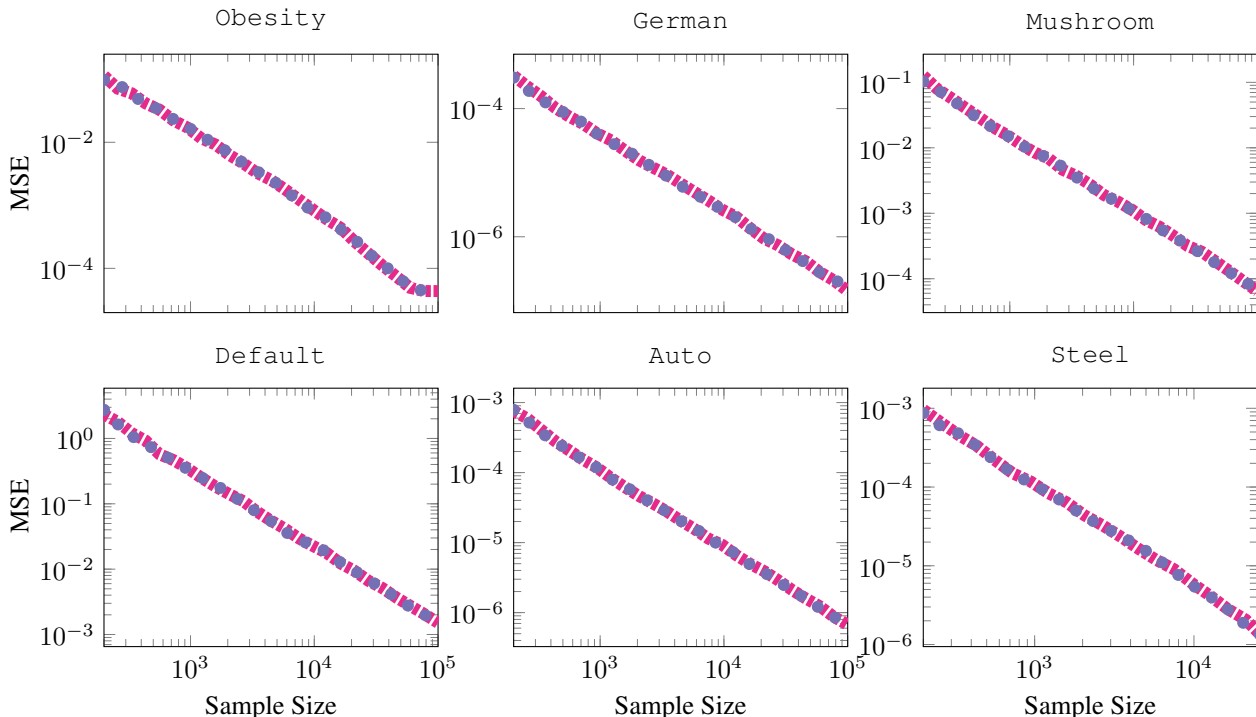

*Figure 13.* **LEVERAGESHAP** •••• **vs. LINEARMSR** ▰▰▰. We report the mean squared error (MSE) of the statistical SHAP estimators across 100 runs as in Figure 12.

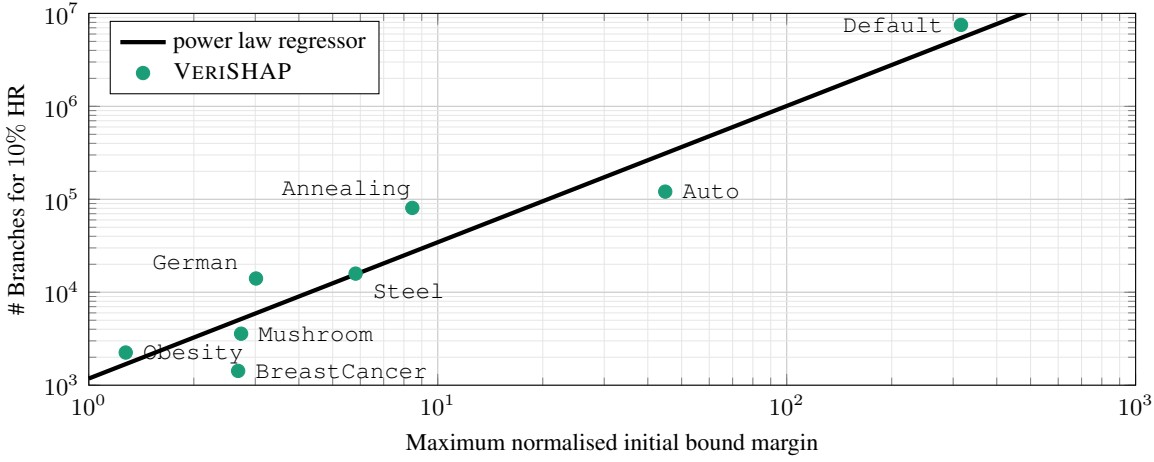

*Figure 14.* **Initial Bounds Half-Range vs. Number of Branchs.** For each tabular network, we plot the largest initial SHAP bounds half-range (HR) $\max_{i\in[n]}(\overline{\varphi_i}^{(1)} - \underline{\varphi_i}^{(1)})/2$ against the number of branches required for VERISHAP to compute SHAP bounds with a maximum half-range of at-most $10\%$ of the network output to attribute ($10\%$ HR). We normalise the initial bounds margin by the network output to attribute to enable the comparison across datasets. A linear regression in log-log space (power law regression) achieves a Pearson correlation coefficient $r$ of $0.956$, demonstrating a strong correlation between the initial bounds half-range and the number of branches required for computing tight bounds.

