# OpenReview forum: "Verified SHAP: Provable Bounds for Exact Shapley Values of Neural Networks"
_ICML.cc/2026/Conference — ICML 2026 regular_

### Official Review · Reviewer_EyRt · 2026-03-07

**Soundness:** 3
**Presentation:** 3
**Significance:** 3
**Originality:** 4
**Overall Recommendation:** 4
**Confidence:** 3

**Summary:**

This paper proposes VERISHAP, a verification-based branch-and-bound framework, and studies the problem of computing lower and upper bounds for exact SHAP values of neural networks without direct enumeration. The core idea is to partition the $2^n$ search space into branches defined by feature constraints, and to apply neural network verification techniques on a mask-based domain in each branch to bound the SHAP marginal contribution. The paper provides a termination guarantee and explains that early exactness is possible for linear models and some piecewise-linear regimes. In experiments, the method shows that it can compute exact SHAP values or tight bounds on much larger search spaces than an existing model, EXACTSHAP, and also suggests its usefulness as a principled ground-truth mechanism for evaluating SHAP estimators in larger-scale neural network settings.

**Compliance With Llm Reviewing Policy:**

Affirmed.

**Final Justification:**

The paper has clear merit in terms of originality and potential significance, and the rebuttal addressed several of my concerns. I think this work is within the range of papers that could reasonably be accepted.

**Key Questions For Authors:**

1. For the vision result, could you report additional diagnostics such as the total number of explored branches, the pruning ratio, and the evolution of the bound gap over time? These statistics would make it much easier to understand why VERISHAP reaches exactness so quickly in some settings.

2. Could you provide a more systematic analysis of failure cases, especially for datasets or activation settings where VERISHAP struggles to obtain tight bounds even on smaller search spaces? In particular, it would be useful to know which factors appear most responsible, such as architecture, activation function, background choice, or model nonlinearity.

3. Do you have any stronger empirical or theoretical evidence for when early exactness is likely to occur in piecewise-linear networks?

**Limitations:**

I think the limitations discussion would become stronger if it also clarified more concretely when practical failure cases happen, and what diagnostics a user can check in advance to judge that VERISHAP may not work well in a given setting.

**Strengths And Weaknesses:**

**Strengths**

The main strengths of this paper are as follows.

1. Paper Structure

Overall, the paper is well organized. The flow from problem setup to theorems, implementation, and experiments is clear, and the ideas of branch partitioning and weight aggregation are explained reasonably well. I also liked that practical engineering issues such as pruning, batching, and branch selection and splitting are discussed together in one space.

2. Technical Novelty

This paper combines coalition partitioning, branch-and-bound, and bound propagation to reformulate exact SHAP computation as a verification-like problem. This feels fresh because it is not just a small improvement of an existing estimator, but a new way to solve the problem itself. In particular, the way the paper computes the total Shapley weight for each branch in closed form or recursively, so that global SHAP bounds can be aggregated without direct coalition enumeration, is practical.

3. Empirical Message

The paper shows that compared with EXACTSHAP, it can compute exact SHAP values or tight bounds on much larger search spaces, and in particular the MNIST superpixel example visually demonstrates well that useful partial bounds can appear quickly even before full convergence (Fig 3). Furthermore, the part where VERISHAP is used as a ground-truth generator to evaluate the accuracy and variability of SHAP estimators on larger search spaces significantly increases the significance of this work from an XAI benchmarking point of view.

**Weaknesses**

On the other hand, the weaknesses of this paper are as follows.

1. Empirical Diagnostics

The paper does not provide enough diagnostics for its most impressive empirical claims. For example, the result that exact SHAP is computed within tens of seconds for a vision example is very impressive, but it would be much more convincing if the paper also presented more detailed information in the main text, such as how many branches were explored, what percentage of branches were certified as linear or constant regions, how much pruning happened, and how the bound gap decreased over time. Similarly, the paper gives only limited failure-mode analysis about why the method works very well on some datasets or activation functions but has difficulty obtaining tight bounds even on smaller search spaces in other cases.

2. Theory for Efficiency

A stronger theory explaining practical efficiency is missing in the paper. The termination argument itself is relatively standard and convincing, and the linear model special case is also easy to understand. But the more important questions are when the verifier bounds become tight quickly, when early exactness happens often inside a piecewise-linear region, and under what architecture or data regimes the branching complexity becomes small. The paper gives some intuition and empirical explanation about these questions, but more concrete complexity characterization or sufficient conditions are still missing.

---

> ### Author Rebuttal · Authors · 2026-03-30
>
> We thank the reviewer for their thorough evaluation and for the interesting questions about our paper.
>
> > For the vision result, could you report additional diagnostics?
>
> Thank you for suggesting these interesting diagnostics. We ran them for our MNIST experiment (Figure 2), and the results are available here: https://figshare.com/s/c315de1c89d417839d3a. The plots report the branch counts, pruning reasons, and number of branches. Overall, they reveal an interesting pattern: the bounds become tight (reaching 10% HR at iteration 120) before any branches are pruned. Moreover, as the middle plot shows, pruning occurs only when the value function is constant within a branch, meaning the model is certified as linear there. In this experiment, this always happens before the branch is fully split, that is, never at a leaf node. We will include similar plots for the remaining datasets in the final version.
>
>
> > A theory on the practical complexity of the VeriSHAP algorithm
>
>
> As stated in the paper, exact SHAP computation is #P-hard in the worst case. As the reviewer correctly noted, our paper nevertheless provides an initial theoretical explanation for the practical efficiency of our algorithm: in the linear setting, VeriSHAP is exact without any branching, identifying a structural regime in which the bounds that are computed by VeriSHAP become exact. We agree that the text can be more explicit about how this argument extends to general neural networks, and we will clarify this in the final version.
>
> Specifically, this perspective extends naturally to piecewise-linear (or nearly piecewise-linear) neural networks. For such models, the runtime of the VeriSHAP algorithm can be upper-bounded at a high level by $\mathcal{O}(B\cdot T_{BP})$, where $B$ denotes the number of explored branches and $T_{BP}$​ is the cost of bound propagation per branch. The central question is therefore how large $B$ becomes: if the VeriSHAP branching quickly restricts the network to subregions on which the marginal contribution $\Delta_i$ is linear, or sufficiently close to linear for verification to be tight, then $B$ remains small, and VeriSHAP is efficient in practice. Hence, the main practical bottleneck is the piecewise-linear geometry of the induced marginal contribution function $\Delta_i$ over the coalition-mask domain, which determines the number of branches required to reach exact or near-exact regions. This provides a broader theoretical explanation for why VeriSHAP can be practically efficient despite the worst-case #P-hardness of exact SHAP computation.
>
> > Could you provide a more systematic analysis of failure cases?
>
> We thoroughly examined the tabular datasets used in our experiments and found that, together with the search space size, the learning-theoretic model variance may explain why VeriSHAP failed for certain datasets. We estimated the model variance by splitting the training set into two halves, training a model on each, and comparing the models' predictions on the test set. The results presented here, https://figshare.com/s/42a48f6155eb74395ac0, indicate that VeriSHAP failures tend to occur when model variance is high, whereas performance remains strong even for large search spaces when variance is lower. We will include a more thorough version of this experiment in the revised paper.
> We also conducted an experiment showing that our results remain stable across different neural architectures. Specifically, we ran additional experiments on the Mushroom dataset with additional activation functions (tanh and Swish), as well as a ResNet. Additional details on this experiment are provided in our rebuttal for reviewer vnAq. We will include an extended version of this experiment in our revised paper.
>
> | Architecture | 10% HR | 1% HR | 0.1% HR | Exact |
> |---|---|---|---|---|
> | ReLU MLP (cf. Table 1) | 17s | 20s | 24s | 25s |
> | Tanh MLP | 19s | 37s | 61s | 70s |
> | Swish MLP | 18s | 35s | 56s | 70s |
> | ReLU ResNet | 72s | 135s | 188s | 199s |
>
> Finally, to further study when VeriSHAP is more efficient, we trained an MNIST model with certified robust training [1], which promotes local adversarial robustness and tends to encourage more stable ReLUs, resulting in fewer and larger linear regions around the training data. The figure at https://figshare.com/s/515e52b43d9ab879b6d5 compares running VeriSHAP on the standard MNIST model from Figure 2 and the model trained for robustness. The results show that VeriSHAP is substantially more efficient on the robustly trained model, reaching 10% HR after 4 iterations, compared to 120 iterations for the standard model. Additional details are provided in our rebuttal for reviewer yorc.
>
> [1] Towards Stable and Efficient Training of Verifiably Robust Neural Networks. (Zhang et al., ICLR 2020)

---

> > ### Author Rebuttal · Reviewer_EyRt · 2026-04-02
> >
> > Thank you for your detailed response. Your experiments addressed my questions. Overall, I think this paper has potential to be accepted, and I keep my positive evaluation for this paper.

---

### Official Review · Reviewer_yorc · 2026-03-08

**Soundness:** 3
**Presentation:** 3
**Significance:** 3
**Originality:** 4
**Overall Recommendation:** 5
**Confidence:** 3

**Summary:**

This paper proposes VERISHAP, a verification-inspired algorithm to compute certified bounds on SHAP values for neural networks, with the ability to tighten these bounds to arbitrary precision and even recover the exact SHAP values. The core idea is to formulate the SHAP marginal contributions in a way amenable to neural network verification tooling, use bound propagation to bound contributions over subsets represented as masks, and combine this with a branch-and-bound style partitioning strategy to iteratively refine bounds. Empirically, the authors show that VERISHAP can compute exact SHAP values for substantially larger search spaces than enumeration-based exact approaches, and can produce tight bounds even when exact computation is infeasible. They also demonstrate how these bounds can be used as a principled evaluation tool for statistical SHAP estimators on larger search spaces.

**Compliance With Llm Reviewing Policy:**

Affirmed.

**Final Justification:**

Given the consistent reviewer consensus and the paper’s clear technical novelty in bringing verification methods to certified and exact SHAP computation, I support acceptance despite the remaining questions about scalability and usage guidance.

**Key Questions For Authors:**

1. Can the authors characterize which factors most determine whether VERISHAP tightens quickly such as Lipschitz/linearity regions and provide guidance for practitioners?

2. The paper notes some datasets where tight bounds were not obtained even with fewer than 60 features. What common traits do these failures share, and are there mitigation strategies such as splitting heuristics?

3. Which network classes are directly supported by the implementation and by the chosen verifier? How does performance change on non-ReLU settings?

4. When exact SHAP is not computed or only bounds, what is the most defensible way to report estimator error?

**Limitations:**

yes

**Strengths And Weaknesses:**

Strengths

The method is grounded in a clear reduction to verification primitives, and the paper explicitly handles the mismatch between discrete coalitions and real-valued verification by representing subsets as masks and relaxing to continuous boxes. The overall empirical story is consistent with the mechanism as when bound propagation can prove constancy across large coalition regions, the algorithm can avoid enumeration and terminate early.

The paper motivates a concrete gap in SHAP evaluation with the lack of reliable ground truth on realistically sized neural settings and then connects the solution cleanly to modern verification methodology. It also provides a helpful experimental summary and clear takeaways as bounds are informative before termination and can be used to evaluate estimators.

Certified bounds for SHAP on neural networks are potentially impactful for explainability. They provide a principled way to assess approximation error and to audit attribution reliability in settings where exact computation is otherwise impossible. The ability to provide useful partial results early with bounds that reveal attribution patterns well before exact termination is a strong practical feature.

The main novelty is the application of neural network verification machinery to the SHAP computation problem by producing certified bounds and enabling exact recovery in cases that substantially exceed prior exact approaches. This also creates a new evaluation lens for SHAP estimators on higher-dimensional neural settings.

Weaknesses

I did not fully check the theoretical proofs end-to-end, so my soundness assessment relies mainly on the coherence of the construction and experimental validation. In addition, while the approach is positioned as applicable to general neural networks, practical scalability will still depend heavily on verifier tightness and activation choices.

For broader accessibility, it would help to include a short diagnostic checklist with requirements like properties of the value network that make bounds tighten quickly vs remain loose, especially since runtime varies substantially by dataset.

The main contribution is within interpretability. Impact is likely strongest for research and high-assurance applications rather than general SHAP usage at very large feature counts, where the method may still be computationally demanding.

How broadly will this scale across architectures or value functions?

---

> ### Author Rebuttal · Authors · 2026-03-30
>
> We thank the reviewer for their thorough review and the helpful questions that will improve our paper.
>
> > Extending our results to additional neural architectures and value functions
>
> We agree that this is an important point. In principle, our method applies to any neural architecture for which bound propagation can be performed. Our implementation already supports a variety of architectures and activation functions beyond ReLUs. To illustrate this, we ran additional experiments on the Mushroom dataset across several architectures, including the paper’s original architecture with tanh and Swish activations, as well as a larger ResNet with three fully connected layers of the input size and residual connections. We will include further networks in our revised paper.
> | Architecture | 10% HR | 1% HR | 0.1% HR | Exact |
> |---|---|---|---|---|
> | ReLU MLP (cf. Table 1) | 17s | 20s | 24s | 25s |
> | Tanh MLP | 19s | 37s | 61s | 70s |
> | Swish MLP | 18s | 35s | 56s | 70s |
> | ReLU ResNet | 72s | 135s | 188s | 199s |
>
>
> Regarding value functions, we evaluated our method on both the standard baseline SHAP value function and the marginal SHAP value function with 100 sampled points, and observed good scalability in both cases. Since our framework is defined for a general value function, we believe a promising direction for future work is to apply it to alternative value functions, such as conditional-expectation SHAP and other causal or distribution-aligned variants. We will make this direction explicit in the final version.
>
>
>
> > How do factors like linearity/Lipschitz regions affect the complexity?
>
> Models with larger linear regions in the input space are more likely to admit tighter bounds, since branching can more quickly reach subdomains that are linear or approximately linear. As a concrete example, we trained a model using certified robust training [1], which encourages adversarial robustness and tends to create networks with larger linear regions. The figure at https://figshare.com/s/515e52b43d9ab879b6d5 compares running VeriSHAP on the standard MNIST model from Figure 2 and the model trained for robustness. The results show that VeriSHAP is substantially more efficient on the robustly trained model, reaching 10% HR after 4 iterations, compared to 120 iterations for the standard model. This suggest that encouraging local linearity during training improves VeriSHAP’s efficiency. We will include this experiment in the final version.
>
> Although this does not directly relate to linearity or Lipschitzness, we also conducted an additional experiment that provides complementary empirical evidence for the same complexity picture. In particular, it shows that the *initial bound* at the root node, obtained via a single inexpensive bound-propagation step, is already predictive of the number of branches required and, hence, VeriSHAP’s runtime for reaching 10% HR. As shown at https://figshare.com/s/de51ac5cec907e9591f9, the number of branches required for reaching 10% HR increases with the initial bound margin. Since linearity and Lipschitzness affect how tight and stable such bounds are, this experiment is consistent with our previous explanations and results. We will include this experiment in our final version as well.
>
> > What common traits are in datasets and models for which VeriSHAP is efficient or less efficient?
>
> This is an interesting point. We thoroughly examined the tabular datasets used in our experiments and found that, together with the search space size, the learning-theoretic model variance may explain why VeriSHAP failed for certain datasets. We estimated the model variance by splitting the training set into two halves, training a model on each, and comparing the models' predictions on the test set. The results presented here, https://figshare.com/s/42a48f6155eb74395ac0, indicate that VeriSHAP failures tend to occur when model variance is high, whereas performance remains strong even for large search spaces when variance is lower. We will include a more thorough version of this experiment in the revised paper. Exploring further splitting heuristics is a promising direction for future research that we will mention in the revised version.
>
> > When the bounds of SHAP are computed, what is the most defensible way to report the estimation error?
>
> A simple metric for comparing estimators is the frequency with which their estimates lie within the SHAP bounds. Beyond this, one can either compute the minimum distance between the estimate and a value in the SHAP bounds (*optimistic* error) or the maximum distance to such a value (*pessimistic* error). Similarly, one can compute an “average” error. While we report optimistic errors in Appendix E.6, all three measurements are defensible.
>
> [1] Towards Stable and Efficient Training of Verifiably Robust Neural Networks. (Zhang et al., ICLR 2020)

---

> > ### Author Rebuttal · Reviewer_yorc · 2026-03-31
> >
> > Thank you for answering all of our questions! We believe that our rating of a 5 is accurate.

---

### Official Review · Reviewer_vnAq · 2026-03-09

**Soundness:** 3
**Presentation:** 3
**Significance:** 2
**Originality:** 3
**Overall Recommendation:** 4
**Confidence:** 3

**Summary:**

This paper proposes VERISHAP, which introduces bound propagation and a branch-and-bound framework from neural network verification into exact SHAP computation, and develops several optimization strategies around this core idea. The experimental results show that, on several tasks, the method can handle larger problem sizes than existing exact methods and represents a step forward in provable SHAP computation.

**Compliance With Llm Reviewing Policy:**

Affirmed.

**Final Justification:**

The authors addressed most of my concerns, I'd like to maintain my score.

**Key Questions For Authors:**

1. Could the authors further break down the sources of error in the final bounds, for example by distinguishing the error introduced by the continuous relaxation from the error caused by the looseness of the verifier itself?
2. The paper demonstrates results on fully connected networks and standard convolutional networks. How well is the method expected to scale to larger and more modern architectures, such as deeper CNNs, residual networks, or transformers?
3. The paper already explains the complementarity between VERISHAP and sampling-based methods. Could the authors further provide clearer practical guidance, for example on what input dimension, error requirement, and compute budget VERISHAP is more suitable for?

**Limitations:**

yes

**Strengths And Weaknesses:**

## Strengths
1. The topic is meaningful. The paper connects verification with SHAP computation, and this perspective is relatively novel.
2. The method section is fairly complete. It includes not only algorithm design, but also theoretical analysis, implementation details, and experimental validation.
3. The experiments are relatively thorough. The comparisons across different verifiers and different search strategies help readers understand the practical behavior of the method.

## Weaknesses
1. The method does not change the fundamentally high complexity of exact SHAP computation itself. Its advantage is therefore mainly in expanding the practically solvable range, rather than fundamentally resolving the difficulty of exact SHAP computation.
2. The paper demonstrates the effect of tight bounds, but the analysis of why the bounds are still not tight enough could be more detailed. For example, it is still unclear how much error comes from the continuous relaxation itself and how much comes from the looseness of the verifier bounds.
3. The experiments mainly cover fully connected networks and standard CNNs, so the practical applicability of VERISHAP to architectures such as ResNets and transformers is still unclear.

---

> ### Author Rebuttal · Authors · 2026-03-30
>
> We thank the reviewer for their helpful and constructive feedback and for acknowledging the importance of our work.
> > How much error comes from the continuous relaxation vs. the looseness of the verifier bounds?
>
> This is an excellent point. We ran an experiment on three benchmarks to examine this explicitly. The results are available at https://figshare.com/s/30fcf678721e02f85c77. Specifically, we compare:
> 1. the initial bounds produced by our method using CROWN-IBP (“Half range of CROWN-IBP Bounds”);
> 2. the *exact* discrete bounds obtained by brute-force enumeration over the discrete $\\{0, 1\\}^n$ space (“Half range of value function over coalitions”); and
> 3. the value function ranges over the continuous relaxation obtained by running a full neural network verification on ${[0, 1]}^n$ (“Half range of value function over continuous relaxation”).
>
> The results show that for the Adult and Obesity datasets, the error is driven mainly by the CROWN bounds rather than by the continuous relaxation. For the German dataset, the continuous relaxation contributes slightly more, but the bounds are already relatively tight here. We hypothesize that this occurs because the German model is relatively smaller, suggesting that the verifier bound looseness may grow with model size.
> > The algorithm, despite its practical significance, does not address the fundamental worst-case complexity of the problem
>
> We agree. As noted in the paper, exactly computing SHAP values is #P-hard, so bringing down the worst-case complexity of the problem to polynomial time would prove P = NP. Still, worst-case hardness does not imply that the problem cannot be solved exactly in practice over trained neural networks. As acknowledged in the review, we significantly advance the *practical* scalability of computing exact SHAP for neural networks.
> > Additional practical guidance on choosing VeriSHAP over other estimators
>
> In safety-critical settings where exact SHAP values or rigorous bounds are needed, VeriSHAP is especially valuable: it is the first method to provide worst-case guarantees for SHAP values while also enabling exact computation over larger search spaces. It is also useful as a ground-truth reference for evaluating SHAP estimators. Naturally, VeriSHAP can take longer to run, so it is most suitable when longer computation is acceptable in exchange for provable correctness.
>
> Regarding the effect of search-space size, required precision, and compute on scalability, we investigate the scalability in search space size in Table 1. In terms of computational cost, all experiments were run on a single NVIDIA L40S GPU with 48GB, which is modest by modern ML standards. Interestingly, Figure 4 suggests that SHAP estimators require *more computation* than VeriSHAP for their average output to fall within the provable precision interval that VeriSHAP attains and certifies. We provide an additional plot https://figshare.com/s/6cbf04fd4e271ac81ce8 highlighting this, which is discussed in our response to reviewer ij6A. Importantly, VeriSHAP can be stopped early once the attained tightness is sufficient for the user’s needs.
>
> To provide practitioners with additional guidance on VeriSHAP’s scalability before running it, we ran an additional experiment showing that the *initial bound* at the root branch, which requires only a single inexpensive bound-propagation step, already provides a useful estimate of the number of branches, and thus of VeriSHAP’s runtime, required to reach tight bounds (10% HR). The figure at https://figshare.com/s/de51ac5cec907e9591f9 shows the number of branches required to reach 10% HR as a function of the initial bound margin. It reveals a clear trend: as the initial bound margin increases, so does the number of branches, and hence the runtime. This suggests that this inexpensive computation can help assess VeriSHAP’s scalability before running the full procedure. The figure at https://figshare.com/s/42a48f6155eb74395ac0 shows that a cheap variance estimate under dataset changes helps predict VeriSHAP’s performance before full computation, as discussed in our response to reviewer EyRt.
> > Does the approach work for additional types of neural architectures?
>
> Yes. Our method applies to any neural architecture for which bound propagation is possible. Our implementation already supports a variety of architectures and activation functions beyond ReLUs. To illustrate this, we ran additional experiments on the Mushroom dataset across several architectures, including the paper’s original architecture with tanh and Swish activations, as well as a larger ResNet with three fully connected layers of the input size and residual connections. We will include further networks in our revised paper.
>
> | Architecture | 10% HR | 1% HR | 0.1% HR | Exact |
> |---|---|---|---|---|
> | ReLU MLP (cf. Table 1) | 17s | 20s | 24s | 25s |
> | Tanh MLP | 19s | 37s | 61s | 70s |
> | Swish MLP | 18s | 35s | 56s | 70s |
> | ReLU ResNet | 72s | 135s | 188s | 199s |

---

> > ### Author Rebuttal · Reviewer_vnAq · 2026-04-01
> >
> > I thank the authors for their detailed response, which has addressed most of my concerns. I therefore maintain my positive evaluation of the paper.

---

### Official Review · Reviewer_ij6A · 2026-03-10

**Soundness:** 3
**Presentation:** 2
**Significance:** 3
**Originality:** 4
**Overall Recommendation:** 5
**Confidence:** 3

**Summary:**

This paper considers the problem of computing Shapley value attributions in the model-specific case of neural networks, specifically ReLU networks. The core idea is to exploit propagated bounds on the network output obtained from neural network verification literature. Using these, bounds on the marginal contributions of the Shapley value are established, which eventually yield bounds on the Shapley values itself. The authors propose an algorithm that recursively splits the feature set into partitions for which the bounds are obtained, and collectively used for bounding the overall Shapley value estimates. The authors showcase their method on multiple benchmarks by demonstrating more efficient performance over Exact SHAP computation from the shap package. On three datasets, where ground-truth computation is infeasible, the authors provide a comparison with Shapley value approximations, where it is shown that the approximators require around $\approx 10^5$ evaluations to achieve comparable performance.

**Compliance With Llm Reviewing Policy:**

Affirmed.

**Final Justification:**

I think this paper provides an elegant way to compute Shapley values for (ReLU) neural networks, and approximations for other structures. The authors addressed my concerns and notational issues during the rebuttal and I there recommend acceptance of the paper.

**Key Questions For Authors:**

- Could you explain more carefully, whether the bounds are not computed by considering a boolean-cube input $[0,1]^n$ or the baseline-input cube $[b,x]^n$, which arises from the baseline / marginal SHAP computation?

**Limitations:**

The authors provide a limitation section, but this section could be improved by limitations such as scalability and applicability.

**Strengths And Weaknesses:**

**Overall assessment**: This paper provides first evidence that internal model-architectures of neural networks can be exploited to speed-up the computation of SHAP explanations. Combining tools from neural network verification and SHAP explanations is an intriguing and novel approach. The main drawback of the proposed method is that convergence can be very slow, which is highlighted by the relatively small datasets (all but one <40 features). In higher dimensions the main competitors are the Shapley value approximators, like KernelSHAP and TreeMSR. The empirical analysis showcases some initial results, but this analysis should have been carried out in more detail to enable practitioners a better understanding which method should be chosen in which case (only three datasets are used for this comparison). While addressing these empirical limitations would substantially improve the paper, I still think the paper provides interesting and novel insights. Provided that my questions/concerns are meaningfully addressed, I am leaning towards acceptance of this paper.

**Soundness**: The methodology and results seem plausible, although I am not an expert in the field of neural network verification. I have some remaining questions regarding the choice of value function and propagation of bounds (see below).

**Presentation**: I think the paper is well structured and presented, and the figures are nicely illustrated, although sometimes they feel a bit too polished / less scientific with many large symbols and colors but they clearly convey the message. A main concern regarding presentation is the connection to neural network verification. The Shapley values are based on a function with binary inputs, whereas the network verification considers intervals in $\mathbb{R}^n$. I assume that the authors wanted to express that the interval is defined by the hypercube spanned by the imputations of a current point to explain $x$ and a chosen reference point $b$ (Baseline SHAP, interval [b,x] in the paper's notation), which is then repeated for the 100 background points. However, as it is currently written (line 200ff, right) it seems that the authors use inputs across the interior of the unit hypercube $[0,1]^{n-1}$ instead. This could have been clarified better. In the algorithm only the set function (value function) is required, and this seems to be independent of the neural network architecture. I understand that the algorithm requires a neural network for the verification bounds but as it reads now, it seems that the algorithm could be applied to any set function v.

**Significance**: This paper studies an important and open problem of model-specific computation of SHAP values in neural network architectures. While this has been done for tree-based models and other architectures, neural networks remain an open question. While this is an interesting contribution itself, the paper does not sufficiently compare with baseline approximation methods of the SHAP values, which would be the main competitor in high-dimensional settings. In its current form, it remains unclear in which cases one should use VeriSHAP and in which cases approximation methods like KernelSHAP. Overall, VeriSHAP is itself could be viewed as an approximation method (obtained from the bounds) until it converges.

**Originality**: This paper proposes an intriguing and novel idea to combine insights from neural network verification to SHAP computation for neural networks. It provides a complementary approach to sampling-based approximation by using the range of the established bounds.

---

> ### Author Rebuttal · Authors · 2026-03-30
>
> We thank the reviewer for their helpful and constructive feedback and for acknowledging the significance of our work.
> > Are the bounds computed on the Boolean hypercube or the baseline-input hypercube?
>
> We agree that this point requires further clarification. The hypercube $[0,1]^{n-1}$ represents the *mask space*, not the input space: under the baseline value function, $\mathbf{v}(\mathbf{m})$ maps a mask $\mathbf{m} \in [0,1]^{n-1}$ to the $[b,x]$ space via $\mathbf{m} * x + (1-\mathbf{m}) * b$, where $*$ is elementwise multiplication. We will add this example to Section 4 to clarify the role of masks, complementing the discussion in Appendix C (lines 1087–1095). For the marginal baseline value function, the computation amounts to summing the upper and lower bounds over all background samples.
> > Comparing VeriSHAP with SHAP estimators across more than three datasets
>
>
> We thank the reviewer for this suggestion. Including additional datasets will strengthen our contribution and better emphasise the significance of VeriSHAP compared to SHAP estimators. We provide a comparison of VeriSHAP with SHAP estimators on the Obesity and German datasets at https://figshare.com/s/5d37fc66a93fae10e81d, complementing the three datasets from Figure 4.
> > A better understanding of when to use VeriSHAP over other SHAP estimators
>
>
> In safety-critical settings where exact SHAP values or rigorous bounds are needed, VeriSHAP is especially valuable: it is the first method to provide worst-case guarantees for SHAP values while also enabling exact computation over larger search spaces. It is also useful as a ground-truth reference for evaluating SHAP estimators. Naturally, VeriSHAP can take longer to run, so it is most suitable when longer computation is acceptable in exchange for provable correctness.
>
> To illustrate that VeriSHAP and SHAP estimators complement each other, we provide an additional plot at https://figshare.com/s/6cbf04fd4e271ac81ce8 that shows how the VeriSHAP bounds and the SHAP estimates from Figure 4 evolve for an increasing runtime budget. As the plot shows, SHAP estimators are faster to compute but do not achieve the precision of VeriSHAP within a comparable runtime budget. It remains important to emphasise that while VeriSHAP provides *provable* bounds that are guaranteed to contain the exact SHAP value, the SHAP estimator ranges are purely empirical and do not provide guarantees.
> > The limitations section can be expanded
>
> We agree and will expand our limitations section in the final version with additional insights from the experiments we conducted for the rebuttal, further clarifying when VeriSHAP is successful and how its runtime can be estimated in advance. For more details, we refer to our answers to the other reviewers.
> > The VeriSHAP algorithm can be seen as an approximation until it converges to the exact value
>
> We agree that VeriSHAP could be viewed as an approximation before termination. However, unlike SHAP estimators, VeriSHAP provides *provable bounds* on the true SHAP values throughout the computation, which are valuable for evaluating SHAP estimators and interpretability in high-stakes domains. To the best of our knowledge, VeriSHAP is the first method to provide worst-case non-asymptotic provable bounds on SHAP values. We view this as an additional significant advantage of VeriSHAP beyond its greater efficiency in computing exact SHAP values.
> > In the algorithm only the set function (value function) is required, and this seems to be independent of the neural network architecture.
>
> While indeed, the bounds in our paper are defined over the value function rather than over the model directly, the SHAP value function itself is defined in terms of the model. Concretely, $v(S) = f(x_S; b_{\bar{S}})$ in the baseline case and $v(S) = \mathbb{E}\_{b \sim \mathcal{D}}[f(x\_S; b\_{\bar{S}})]$  in the marginal case. Therefore, bounding the value function requires bounding the model $f$ itself, which we do via (recursive) bound propagation over $v$.
>
>
> In principle, our method could extend to other ML models, provided suitable model bounds can be computed. This may be possible for models such as non-linear kernel methods, but we are not aware of existing bound-propagation techniques for them. Our algorithm builds on recent progress in neural network bound propagation from the verification literature, and similar advances for other model classes could further extend its applicability. To demonstrate generalisation across other *neural* architectures, we ran VeriSHAP on additional neural networks trained on the Mushroom dataset.
>
> | Architecture | 10% HR | 1% HR | 0.1% HR | Exact |
> |---|---|---|---|---|
> | ReLU MLP (cf. Table 1) | 17s | 20s | 24s | 25s |
> | Tanh MLP | 19s | 37s | 61s | 70s |
> | Swish MLP | 18s | 35s | 56s | 70s |
> | ReLU ResNet | 72s | 135s | 188s | 199s |
>
>
> Additional details on this experiment are provided in our rebuttal for reviewer vnAq. We will extend these experiments for the revised paper.

---

> > ### Author Rebuttal · Reviewer_ij6A · 2026-04-01
> >
> > I thank the authors for addressing my concerns and provide further clarifications and experiments in the main paper. I think this improved the paper and contribution, and I will therefore raise my score.

---

### Decision · Program_Chairs · 2026-04-30

**Decision:**

Accept (regular)

**Comment:**

The paper studies an important concept in explainable AI, whether one can obtain provable and eventually exact SHAP values for neural networks by importing tools from neural network verification. This article aims to present a pressing question for the community, namely how to move beyond heuristic or purely sampling-based feature attributions when exact computation is combinatorial and often infeasible.

The reviewer discussion shows a fairly clear positive consensus, and the remaining disagreement is mainly about scope, practical guidance, and how broadly the current empirical evidence extends beyond the architectures studied. The paper’s core contribution is to introduce VERISHAP, a branch-and-bound framework that uses verification-style bound propagation to compute lower and upper bounds on SHAP values, tighten them, and in finite time recover the exact value. Empirically, the paper does support the claim that VERISHAP expands the practically solvable range compared with exact enumeration-based methods, and the MNIST example plus the tabular timing table are compelling evidence that the method can produce useful certified intervals before exact convergence. At the same time, I agree with reviewers that the paper should be more explicit that this is a first step, not a general solution to exact SHAP at scale; the rebuttal appropriately acknowledges that worst-case complexity is still exponential and that practicality depends on geometry/linearity of the induced mask-domain function.

For the camera-ready, I strongly encourage the authors to make the edits from reviewer discussion, including adding a discussion on the relaxed function being verified on mask space, breakdown of the two error sources, discussion comparing with KernelSHAP/TreeMSR-style estimators, and the additional neural architectures in the rebuttal. Also please proofread the manuscript carefully to remove any typos, e.g. in proof of Proposition 3.2, the $k+1$ term should be $k+r$.